# DISCOVERING A SET OF POLICIES FOR THE WORST CASE REWARD

**Tom Zahavy,**[*] **Andre Barreto, Daniel J Mankowitz, Shaobo Hou, Brendan O'Donoghue, Iurii Kemaev and Satinder Singh**
DeepMind

## ABSTRACT

We study the problem of how to construct a set of policies that can be composed together to solve a collection of reinforcement learning tasks. Each task is a different reward function defined as a linear combination of known features. We consider a specific class of policy compositions which we call set improving policies (SIPs): given a set of policies and a set of tasks, a SIP is any composition of the former whose performance is at least as good as that of its constituents across all the tasks. We focus on the most conservative instantiation of SIPs, set-max policies (SMPs), so our analysis extends to any SIP. This includes known policy-composition operators like generalized policy improvement. Our main contribution is a policy iteration algorithm that builds a set of policies in order to maximize the worst-case performance of the resulting SMP on the set of tasks. The algorithm works by successively adding new policies to the set. We show that the worst-case performance of the resulting SMP strictly improves at each iteration, and the algorithm only stops when there does not exist a policy that leads to improved performance. We empirically evaluate our algorithm on a grid world and also on a set of domains from the DeepMind control suite. We confirm our theoretical results regarding the monotonically improving performance of our algorithm. Interestingly, we also show empirically that the sets of policies computed by the algorithm are diverse, leading to different trajectories in the grid world and very distinct locomotion skills in the control suite.

## 1 INTRODUCTION

Reinforcement learning (RL) is concerned with building agents that can learn to act so as to maximize reward through trial-and-error interaction with the environment. There are several reasons why it can be useful for an agent to learn about multiple ways of behaving, *i.e.*, learn about multiple policies. The agent may want to achieve multiple tasks (or subgoals) in a lifelong learning setting and may learn a separate policy for each task, reusing them as needed when tasks reoccur. The agent may have a hierarchical architecture in which many policies are learned at a lower level while an upper level policy learns to combine them in useful ways, such as to accelerate learning on a single task or to transfer efficiently to a new task. Learning about multiple policies in the form of options (Sutton et al., 1999a) can be a good way to achieve temporal abstraction; again this can be used to quickly plan good policies for new tasks. In this paper we abstract away from these specific scenarios and ask the following question: what set of policies should the agent pre-learn in order to guarantee good performance under the worst-case reward? A satisfactory answer to this question could be useful in all the scenarios discussed above and potentially many others.

There are two components to the question above: $(i)$ what policies should be in the set, and $(ii)$ how to compose a policy to be used on a new task from the policies in the set. To answer $(ii)$, we propose the concept of a *set improving policy* (SIP). Given any set of $n$ policies, a SIP is any composition of these policies whose performance is at least as good as, and generally better than, that of all of the constituent policies in the set. We present two policy composition (or improvement) operators that lead to a SIP. The first is called *set-max policy* (SMP). Given a distribution over states, a SMP chooses from $n$ policies the one that leads to the highest expected value. The second SIP operator is *generalized policy improvement* (Barreto et al., 2017, GPI). Given a set of $n$ policies and their associated action-value functions, GPI is a natural extension of regular policy improvement in which the agent acts greedily in each state with respect to the maximum over the set of action-values

---

[*]`tomzahavy@google.com`

functions. Although SMP provides weaker guarantees than GPI (we will show this below), it is more amenable to analysis and thus we will use it exclusively for our theoretical results. However, since SMP's performance serve as a lower bound to GPI's, the results we derive for the former also apply to the latter. In our illustrative experiments we will show this result empirically.

Now that we have fixed the answer to $(ii)$, *i.e.*, how to compose pre-learned policies for a new reward function, we can leverage it to address $(i)$: what criterion to use to pre-learn the policies. Here, one can appeal to heuristics such as the ones advocating that the set of pre-learned policies should be as diverse as possible (Eysenbach et al., 2018; Gregor et al., 2016; Grimm et al., 2019; Hansen et al., 2019). In this paper we will use the formal criterion of *robustness*, *i.e.*, we will seek a set of policies that do as well as possible in the worst-case scenario. Thus, the problem of interest to this paper is as follows: how to define and discover a set of $n$ policies that maximize the worst possible performance of the resulting SMP across all possible tasks? Interestingly, as we will discuss, the solution to this robustness problem naturally leads to a diverse set of policies.

To solve the problem posed above we make two assumptions: (A1) that tasks differ only in their reward functions, and (A2) that reward functions are linear combinations of known features. These two assumptions allow us to leverage the concept of *successor features* (SFs) and work in apprenticeship learning. As our main contribution in this paper, we present an algorithm that iteratively builds a set of policies such that SMP's performance with respect to the worst case reward provably improves in each iteration, stopping when no such greedy improvement is possible. We also provide a closed-form expression to compute the worst-case performance of our algorithm at each iteration. This means that, given tasks satisfying Assumptions A1 and A2, we are able to provably construct a SIP that can quickly adapt to any task with guaranteed worst-case performance.

**Related Work.** The proposed approach has interesting connections with hierarchical RL (HRL) (Sutton et al., 1999b; Dietterich, 2000). We can think of SMP (and GPI) as a higher-level policy-selection mechanism that is fixed *a priori*. Under this interpretation, the problem we are solving can be seen as the definition and discovery of lower-level policies that will lead to a robust hierarchical agent.

There are interesting parallels between robustness and diversity. For example, diverse stock portfolios have less risk. In robust least squares (El Ghaoui & Lebret, 1997; Xu et al., 2009), the goal is to find a solution that will perform well with respect to (w.r.t) data perturbations. This leads to a min-max formulation, and there are known equivalences between solving a robust (min-max) problem and the diversity of the solution (via regularization) (Xu & Mannor, 2012). Our work is also related to robust Markov decision processes (MDPs) (Nilim & El Ghaoui, 2005), but our focus is on a different aspect of the problem. While in robust MDPs the uncertainty is w.r.t the dynamics of the environment, here we focus on uncertainty w.r.t the reward and assume that the dynamics are fixed. More importantly, we are interested in the hierarchical aspect of the problem – how to discover and compose a set of policies. In contrast, solutions to robust MDPs are typically composed of a single policy.

In Apprenticeship Learning (AL; Abbeel & Ng, 2004) the goal is also to solve a min-max problem in which the agent is expected to perform as well as an expert w.r.t any reward. If we ignore the expert, AL algorithms can be used to find a single policy that performs well w.r.t any reward. The solution to this problem (when there is no expert) is the policy whose SFs have the smallest possible norm. When the SFs are in the simplex (as in tabular MDPs) the vector with the smallest $\ell_2$ norm puts equal probabilities on its coordinates, and is therefore "diverse" (making an equivalence between the robust min-max formulation and the diversity perspective). In that sense, our problem can be seen as a modified AL setup where: (a) no expert demonstrations are available (b) the agent is allowed to observe the reward at test time, and (c) the goal is to learn a set of constituent policies.

## 2 PRELIMINARIES

We will model our problem of interest using a family of Markov Decision Processes (MDPs). An MDP is a tuple $M \triangleq (S, A, P, r, \gamma, D)$, where $S$ is the set of states, $A$ is the set of actions, $P = \{P^a \mid a \in A\}$ is the set of transition kernels, $\gamma \in [0, 1]$ is the discount factor and $D$ is the initial state distribution. The function $r : S \times A \times S \mapsto \mathbb{R}$ defines the rewards, and thus the agent's objective; here we are interested in multiple reward functions, as we explain next.

Let $\boldsymbol{\phi}(s, a, s') \in [0, 1]^d$ be an observable vector of features (our analysis only requires the features to be bounded; we use $[0, 1]$ for ease of exposition). We are interested in the set of tasks induced by all possible linear combinations of the features $\boldsymbol{\phi}$. Specifically, for any $\mathbf{w} \in \mathbb{R}^d$, we can define a reward function $r_{\mathbf{w}}(s, a, s') = \mathbf{w} \cdot \boldsymbol{\phi}(s, a, s')$. Given $\mathbf{w}$, the reward $r_{\mathbf{w}}$ is well defined and we will use the terms $\mathbf{w}$ and $r_{\mathbf{w}}$ interchangeably to refer to the RL task induced by it. Formally, we are interested in

the following set of MDPs:

$$\mathcal{M}_\phi \triangleq \{(S, A, P, r_\mathbf{w}, \gamma, D) \,|\, \mathbf{w} \in \mathcal{W}\}. \tag{1}$$

In general, $\mathcal{W}$ is any convex set, but we will focus on the $\ell_2$ $d$-dimensional ball denoted by $\mathcal{W} = \mathcal{B}_2$. This choice is not restricting, since the optimal policy in an MDP is invariant with respect to the scale of the rewards and the $\ell_2$ ball contains all the directions.

A policy in an MDP $M \in \mathcal{M}_\phi$, denoted by $\pi \in \Pi$, is a mapping $\pi : S \to \mathcal{P}(A)$, where $\mathcal{P}(A)$ is the space of probability distributions over $A$. For a policy $\pi$ we define the *successor features* (SFs) as

$$\boldsymbol{\psi}^\pi(s, a) \triangleq (1 - \gamma) \cdot \mathbb{E}\Big[\sum_{t=0}^\infty \gamma^t \boldsymbol{\phi}(s_t, a_t, s_{t+1}) | P, \pi, s_t = s, a_t = a\Big]. \tag{2}$$

The multiplication by $1 - \gamma$ together with the fact that the features $\phi$ are in $[0, 1]$ assures that $\boldsymbol{\psi}^\pi(s, a) \in [0, 1]^d$ for all $(s, a) \in S \times A$.[1] We also define SFs that are conditioned on the initial state distribution $D$ and the policy $\pi$ as: $\boldsymbol{\psi}^\pi \triangleq \mathbb{E}[\boldsymbol{\psi}^\pi(s, a)|D, \pi] = \mathbb{E}_{s \sim D, a \sim \pi(s)}\boldsymbol{\psi}^\pi(s, a)$. It should be clear that the SFs are conditioned on $D$ and $\pi$ whenever they are not written as a function of states and actions like in Eq. (2). Note that, given a policy $\pi$, $\boldsymbol{\psi}^\pi$ is simply a vector in $[0, 1]^d$. Since we will be dealing with multiple policies, we will use superscripts to refer to them—that is, we use $\pi^i$ to refer to the $i$-th policy. To keep the notation simple, we will refer to the SFs of policy $\pi^i$ as $\boldsymbol{\psi}^i$. We define the action-value function (or $Q$-function) of policy $\pi$ under reward $r_\mathbf{w}$ as

$$Q_\mathbf{w}^\pi(s, a) \triangleq (1 - \gamma)\mathbb{E}\Big[\sum_{t=0}^\infty \gamma^t \boldsymbol{\phi}(s_t, a_t, s_{t+1}) \cdot \mathbf{w}|P, \pi, s_t = s, a_t = a\Big] = \boldsymbol{\psi}^\pi(s, a) \cdot \mathbf{w}.$$

We define the value of a policy $\pi$ as $v_\mathbf{w}^\pi \triangleq (1 - \gamma)\mathbb{E}\Big[\sum_{t=0}^\infty \gamma^t \mathbf{w} \cdot \phi(s_t)|\pi, P, D\Big] = \psi^\pi \cdot \mathbf{w}$. Note that $v_\mathbf{w}^\pi$ is a scalar, corresponding to the expected value of policy $\pi$ under the initial state distribution $D$, given by

$$v_\mathbf{w}^\pi = \mathbb{E}[Q_\mathbf{w}^\pi(s, a)|D, \pi] = \mathbb{E}_{s \sim D, a \sim \pi(s)}Q_\mathbf{w}^\pi(s, a). \tag{3}$$

## 3 COMPOSING POLICIES TO SOLVE A SET OF MDPs

As described, we are interested in solving all the tasks $\mathbf{w} \in \mathcal{W}$ in the set of MDPs $\mathcal{M}_\phi$ defined in (1). We will approach this problem by learning policies associated with specific rewards $\mathbf{w}$ and then composing them to build a higher-level policy that performs well across all the tasks. We call this higher-level policy a *generalized policy*, defined as (Barreto et al., 2020):

**Definition 1** (Generalized policy). *Given a set of MDPs $\mathcal{M}_\phi$, a generalized policy is a function $\pi : S \times \mathcal{W} \mapsto \mathcal{P}(A)$ that maps a state $s$ and a task $\mathbf{w}$ onto a distribution over actions.*

We can think of a generalized policy as a regular policy parameterized by a task, since for a fixed $\mathbf{w}$ we have $\pi(\cdot; \mathbf{w}) : S \mapsto \mathbb{P}(A)$. We now focus our attention on a specific class of generalized policies that are composed of other policies:

**Definition 2** (SIP). *Given a set of MDPs $\mathcal{M}_\phi$ and a set of $n$ policies $\Pi^n = \{\pi^i\}_{i=1}^n$, a set improving policy (SIP) $\pi^{SIP}$ is any generalized policy such that:*

$$v_{\Pi^n, \mathbf{w}}^{SIP} \geq v_\mathbf{w}^i \text{ for all } \pi^i \in \Pi^n \text{ and all } \mathbf{w} \in \mathcal{W}, \tag{4}$$

*where $v_{\Pi^n, \mathbf{w}}^{SIP}$ and $v_\mathbf{w}^i$ are the value functions of $\pi_{\Pi^n}^{SIP}(\cdot; \mathbf{w})$ and the policies $\pi^i \in \Pi^n$ under reward $r_\mathbf{w}$.*

We have been deliberately vague about the specific way the policies $\pi^i \in \Pi^n$ are combined to form a SIP to have as inclusive a concept as possible. We now describe two concrete ways to construct a SIP.

**Definition 3** (SMP). *Let $\Pi^n = \{\pi^i\}_{i=1}^n$ be a set of $n$ policies and let $v^i$ be the corresponding value functions defined analogously to (3) for an arbitrary reward. A set-max policy (SMP) is defined as*

$$\pi_{\Pi^n}^{SMP}(s; \mathbf{w}) = \pi^k(s), \text{ with } k = \arg\max_{i \in [1, ..., n]} v_\mathbf{w}^i.$$

---

[1]While we focus on the most common, discounted RL criteria, all of our results will hold in the finite horizon and average reward criteria (see, for example, Puterman (1984)). Concretely, in these scenarios there exist normalizations for the SFs whose effect are equivalent to that of the multiplication by $1 - \gamma$. In the finite-horizon case we can simply multiply the SFs by $1/H$. In the average reward case, there is no multiplication (Zahavy et al., 2020b) and the value function is measured under the stationary distribution (instead of $D$).

Combining the concepts of SMP and SFs we can build a SIP for $\mathcal{M}_\phi$. Given the SFs of the policies $\pi^i \in \Pi^n$, $\{\psi^i\}_{i=1}^n$, we can quickly compute a generalized SMP as

$$\pi_{\Pi^n}^{\text{SMP}}(s; \mathbf{w}) = \pi^k(s), \text{ with } k = \arg \max_{i \in [1,\ldots,n]} \{\mathbf{w} \cdot \psi^i\}. \tag{5}$$

Since the value of a SMP under reward $\mathbf{w}$ is given by $v_{\Pi^n, \mathbf{w}}^{\text{SMP}} = \max_{i \in [1,\ldots,n]} v_{\mathbf{w}}^i$, it trivially qualifies as a SIP as per Definition 2. In fact, the generalized policy $\pi_{\Pi^n}^{\text{SMP}}$ defined in (5) is in some sense the most conservative SIP possible, as it will always satisfy (4) with equality. This means that any other SIP will perform at least as well as the SIP induced by SMP. We formalize this notion below:

**Lemma 1.** *Let $\pi_{\Pi^n}^{SMP}$ be a SMP defined as in (5) and let $\pi : S \times \mathcal{W} \mapsto \mathbb{P}(A)$ be any generalized policy. Then, given a set of $n$ policies $\Pi^n$, $\pi$ is a SIP if and only if $v_{\Pi^n, \mathbf{w}}^\pi \geq v_{\Pi^n, \mathbf{w}}^{SMP}$ for all $\mathbf{w} \in \mathcal{W}$.*

Due to space constraints, all the proofs can be found in the supplementary material. Lemma 1 allows us to use SMP to derive results that apply to all SIPs. For example, a lower bound for $v_{\Pi^n, \mathbf{w}}^{\text{SMP}}$ automatically applies to all possible $v_{\Pi^n, \mathbf{w}}^{\text{SIP}}$. Lemma 1 also allows us to treat SMP as a criterion to determine whether a given generalized policy qualifies as a SIP. We illustrate this by introducing a second candidate to construct a SIP called *generalized policy improvement* (Barreto et al., 2017; 2018; 2020, GPI):

**Definition 4** (GPI policy). *Given a set of $n$ policies $\Pi^n = \{\pi^i\}_{i=1}^n$ and corresponding Q-functions $Q_{\mathbf{w}}^i$ computed under an arbitrary reward $\mathbf{w}$, the GPI policy is defined as*

$$\pi_{\Pi^n}^{GPI}(s; \mathbf{w}) = \arg \max_a \max_i Q_{\mathbf{w}}^i(s, a).$$

Again, we can combine GPI and SFs to build a generalized policy. Given the SFs of the policies $\pi^i \in \Pi^n$, $\{\psi^i\}_{i=1}^n$, we can quickly compute the generalized GPI policy as $\pi_{\Pi^n}^{\text{GPI}}(s; \mathbf{w}) = \arg \max_a \max_i \psi^i(s, a) \cdot \mathbf{w}$. Note that the maximization in GPI is performed in each state and uses the $Q$-functions of the constituent policies. In contrast, SMP maximizes over value functions (not $Q$-functions), with an expectation over states taken with respect to the initial state distribution $D$. For this reason, GPI is a stronger composition than SMP. We now formalize this intuition:

**Lemma 2.** *For any reward $\mathbf{w} \in \mathcal{W}$ and any set of policies $\Pi^n$, we have that $v_{\Pi^n, \mathbf{w}}^{GPI} \geq v_{\Pi^n, \mathbf{w}}^{SMP}$.*

Lemma 2 implies that for any set of policies it is always better to use a GPI policy rather than an SMP (as we will confirm in the experiments). As a consequence, it also certifies that the generalized GPI policy $\pi_{\Pi^n}^{\text{GPI}}(s; \mathbf{w})$ qualifies as a SIP (Lemma 1).

We have described two ways of constructing a SIP by combining SMP and GPI with SFs. Other similar strategies might be possible, for example by using local SARSA (Russell & Zimdars, 2003; Sprague & Ballard, 2003) as the basic mechanism to compose a set of value functions. We also note that in some cases it is possible to define a generalized policy (Definition 1), that is not necessarily a SIP (Eq. (5)), but is guaranteed to perform better than any SIP in expectation. For example, a combination of maximization, randomization and local search have been shown to be optimal in expectation among generalized policies in tabular MDPs with collectible rewards (Zahavy et al., 2020c). That said, we note that some compositions of policies that may at first seem like a SIP do not qualify as such. For example, a mixed policy is a linear (convex) combination of policies that assigns probabilities to the policies in the set and samples from them. When the mixed policy is mixing the best policy in the set with a less performant policy then it will result in a policy that is not as good as the best single policy in the set (Zahavy et al., 2020c).

**Problem formulation.** We are now ready to formalize the problem we are interested in. Given a set of MDPs $\mathcal{M}_\phi$, as defined in (1), we want to construct a set of $n$ policies $\Pi^n = \{\pi^i\}_{i=1}^n$, such that the performance of the SMP defined on that set $\pi_{\Pi^n}^{\text{SMP}}$ will have the optimal worst-case performance over all rewards $\mathbf{w} \in \mathcal{W}$. That is, we want to solve the following problem:

$$\arg \max_{\Pi^n \subseteq \Pi} \min_{\mathbf{w}} v_{\Pi^n, \mathbf{w}}^{\text{SMP}}. \tag{6}$$

Note that, since $v_{\Pi^n, \mathbf{w}}^{\text{SMP}} \leq v_{\Pi^n, \mathbf{w}}^{\text{SIP}}$ for any SIP, $\Pi^n$ and $\mathbf{w}$, as shown in Lemma 1, by finding a good set for (6) we are also improving the performance of all SIPs (including GPI).

## 4 AN ITERATIVE METHOD TO CONSTRUCT A SET-MAX POLICY

We now present and analyze an iterative algorithm to solve problem (6). We begin by defining the worst case or adversarial reward associated with the generalized SMP policy:

**Definition 5** (Adversarial reward for an SMP). *Given a set of policies $\Pi^n$, we denote by $\bar{\mathbf{w}}_{\Pi^n}^{SMP} = \arg\min_{\mathbf{w} \in B_2} v_{\Pi^n, \mathbf{w}}^{SMP}$ the worst case reward w.r.t the SMP $\pi_{\Pi^n}^{SMP}$ defined in (5). In addition, the value of the SMP w.r.t to $\bar{\mathbf{w}}_{\Pi^n}^{SMP}$ is defined by $\bar{v}_{\Pi^n}^{SMP} = \min_{\mathbf{w} \in B_2} v_{\Pi^n, \mathbf{w}}^{SMP}$.*

We are interested in finding a set of policies $\Pi^n$ such that the performance of the resulting SMP will be optimal w.r.t its adversarial reward $\bar{\mathbf{w}}_{\Pi^n}^{SMP}$. This leads to a reformulation of (6) as a max-min-max optimization for discovering robust policies:

$$\arg\max_{\Pi^n \subseteq \Pi} \bar{v}_{\Pi^n}^{SMP} = \arg\max_{\Pi^n \subseteq \Pi} \min_{\mathbf{w} \in B_2} v_{\Pi^n, \mathbf{w}}^{SMP} = \arg\max_{\Pi^n \subseteq \Pi} \min_{\mathbf{w} \in B_2} \max_{i \in [1,..,n]} \boldsymbol{\psi}^i \cdot \mathbf{w}. \quad (7)$$

The order in which the maximizations and the minimization are performed in (7) is important. $(i)$ The inner maximization over policies (or SFs), by the SMP, is performed last. This means that, for a fixed set of policies $\Pi^n$ and a fixed reward $\mathbf{w}$, SMP selects the best policy in the set. $(ii)$ The minimization over rewards $\mathbf{w}$ happens second, that is, for a fixed set of policies $\Pi^n$, we compute the value of the generalized SMP $\pi_{\Pi^n}^{SMP}(\cdot; \mathbf{w})$ for any reward $\mathbf{w}$, and then minimize the maximum of these values. $(iii)$ Finally, for any set of policies, there is an associated worst case reward for the SMP, and we are looking for policies that maximize this value.

---

**Algorithm 1** SMP worst case policy iteration

**Initialize:** Sample $\mathbf{w} \sim N(\bar{0}, \bar{1}), \Pi^0 \leftarrow \{\ \}, \pi^1 \leftarrow \arg\max_{\pi \in \Pi} \mathbf{w} \cdot \boldsymbol{\psi}^\pi, t \leftarrow 1$
$\bar{v}_{\Pi^1}^{SMP} \leftarrow -||\boldsymbol{\psi}^1||$
**repeat**
    $\Pi^t \leftarrow \Pi^{t-1} + \{\pi^t\}$
    $\bar{\mathbf{w}}_{\Pi^t}^{SMP} \leftarrow$ solution to (8)
    $\pi^{t+1} \leftarrow$ solution of the RL task $\bar{\mathbf{w}}_{\Pi^t}^{SMP}$
    $t \leftarrow t + 1$
**until** $v_{\bar{\mathbf{w}}_{\Pi^n}^{SMP}}^t \leq \bar{v}_{\Pi^{t-1}}^{SMP}$
**return** $\Pi^{t-1}$

---

The inner maximization $(i)$ is simple: it comes down to computing $n$ dot-products $\boldsymbol{\psi}^i \cdot \mathbf{w}$, $i = 1, 2, \ldots, n$, and comparing the resulting values. The minimization problem $(ii)$ is slightly more complicated, but fortunately easy to solve. To see this, note that this problem can be rewritten as:

$$\bar{\mathbf{w}}_{\Pi^n}^{SMP} = \arg\min_{\mathbf{w} \in \mathbb{B}_2} \max_{i \in [1,...,n]} \{\mathbf{w} \cdot \boldsymbol{\psi}^1, \ldots, \mathbf{w} \cdot \boldsymbol{\psi}^n\}. \ s.t. \ ||\mathbf{w}||^2 - 1 \leq 0. \quad (8)$$

Eq. (8) is a convex optimization problem that can be easily solved using standard techniques, like gradient descent, and off-the-shelf solvers (Diamond & Boyd, 2016; Boyd et al., 2004). We note that the minimizer of Eq. (8) is a function of policy set. As a result, the set forces the worst case reward to make a trade-off – it has to "choose" the coordinates it "wants" to be more adversarial for. This trade-off is what encourages the worst case reward to be diverse across iterations (w.r.t different sets). We note that this property holds since we are optimizing over $\mathbb{B}_2$ but it will not necessary be the case for other convex sets. For example, in the case of $\mathbb{B}_\infty$ the internal minimization problem in the above has a single solution - a vector with -1 in all of its coordinates.

The outer maximization problem $(iii)$ can be difficult to solve if we are searching over all possible sets of policies $\Pi^n \subseteq \Pi$. Instead, we propose an incremental approach in which policies $\pi^i$ are successively added to an initially empty set $\Pi^0$. This is possible because the solution $\bar{\mathbf{w}}_{\Pi^n}^{SMP}$ of (8) gives rise to a well-defined RL problem in which the rewards are given by $r_\mathbf{w}(s, a, s') = \bar{\mathbf{w}}_{\Pi^n}^{SMP} \cdot \boldsymbol{\phi}(s, a, s')$. This problem can be solved using any standard RL algorithm. So, once we have a solution $\bar{\mathbf{w}}_{\Pi^n}^{SMP}$ for (8), we solve the induced RL problem using any algorithm and add the resulting policy $\pi^{n+1}$ to $\Pi^n$ (or, rather, the associated SFs $\boldsymbol{\psi}^{n+1}$).

Algorithm 1 has a step by step description of the proposed method. The algorithm is initialized by adding a policy $\pi^1$ that maximizes a random reward vector $\mathbf{w}$ to the set $\Pi^0$, such that $\Pi^1 = \{\pi^1\}$. At each subsequent iteration $t$ the algorithm computes the worst case reward $\bar{\mathbf{w}}_{\Pi^t}^{SMP}$ w.r.t to the current set $\Pi^t$ by solving (8). The algorithm then finds a policy $\pi^{t+1}$ that solves the task induced by $\bar{\mathbf{w}}_{\Pi^t}^{SMP}$. If the value of $\pi^{t+1}$ w.r.t $\bar{\mathbf{w}}_{\Pi^t}^{SMP}$ is strictly larger than $\bar{v}_{\Pi^t}^{SMP}$ the algorithm continues for another iteration, with $\pi^{t+1}$ added to the set. Otherwise, the algorithm stops. As mentioned before, the set of policies $\Pi^t$ computed by Algorithm 1 can also be used with GPI. The resulting GPI policy will do at least as well as the SMP counterpart on any task $\mathbf{w}$ (Lemma 2); in particular, the GPI's worst-case performance will be lower bounded by $\bar{v}_{\Pi^n}^{SMP}$.

### 4.1 Theoretical Analysis

Algorithm 1 produces a sequence of policy sets $\Pi^1, \Pi^2, \ldots$ The definition of SMP guarantees that enlarging a set of policies always leads to a soft improvement in performance, so $\bar{v}^{\text{SMP}}_{\Pi^{t+1}} \geq \bar{v}^{\text{SMP}}_{\Pi^t} \geq \ldots \geq \bar{v}^{\text{SMP}}_{\{\pi^1\}}$. We now show that the improvement in each iteration of our algorithm is in fact strict.

**Theorem 1** (Strict improvement). *Let $\Pi^1, \ldots, \Pi^t$ be the sets of policies constructed by Algorithm 1. We have that the worst-case performance of the SMP induced by these set is strictly improving in each iteration, that is: $\bar{v}^{\text{SMP}}_{\Pi^{t+1}} > \bar{v}^{\text{SMP}}_{\Pi^t}$. Furthermore, when the algorithm stops, there does not exist a single policy $\pi^{t+1}$ such that adding it to $\Pi^t$ will result in improvement: $\nexists \, \pi^{t+1} \in \Pi$ s.t. $\bar{v}^{\text{SMP}}_{\Pi^t + \{\pi\}} > \bar{v}^{\text{SMP}}_{\Pi^t}$.*

In general we cannot say anything about the value of the SMP returned by Algorithm 1. However, in some special cases we can upper bound it. One such case is when the SFs lie in the simplex.

**Lemma 3** (Impossibility result). *For the special case where the SFs associated with any policy are in the simplex, the value of the SMP w.r.t the worst case reward for any set of policies is less than or equal to $-1/\sqrt{d}$. In addition, there exists an MDP where this upper bound is attainable.*

One example where the SFs are in the simplex is when the features $\phi$ are "one-hot vectors", that is, they only have one nonzero element. This happens for example in a tabular representation, in which case the SFs correspond to stationary state distributions. Another example are the features induced by state aggregation, since these are simple indicator functions associating states to clusters (Singh et al., 1995). We will show in our experiments that when state aggregation is used our algorithm achieves the upper bound of Lemma 3 in practice.

Finally, we observe that not all the policies in the set $\Pi^t$ are needed at each point in time, and we can guarantee strict improvement even if we remove the "inactive" policies from $\Pi^t$, as we show below.

**Definition 6** (Active policies). *Given a set of $n$ policies $\Pi^n$, and an associated worst case reward $\bar{\mathbf{w}}^{SMP}_{\Pi^n}$, the subset of active policies $\Pi_a(\Pi^n)$ are the policies in $\Pi^n$ that achieve $\bar{v}^{SMP}_{\Pi^n}$ w.r.t $\bar{\mathbf{w}}^{SMP}_{\Pi^n}$ : $\Pi_a(\Pi^n) = \left\{ \pi \in \Pi^n : \boldsymbol{\psi}^\pi \cdot \bar{\mathbf{w}}^{SMP}_{\Pi^n} = \bar{v}^{SMP}_{\Pi^n} \right\}$.*

**Theorem 2** (Sufficiency of Active policies). *For any set of policies $\Pi^n$, $\pi^{SMP}_{\Pi_a(\Pi^n)}$ achieves the same value w.r.t the worst case reward as $\pi^{SMP}_{\Pi^n}$, that is, $\bar{v}^{SMP}_{\Pi^n} = \bar{v}^{SMP}_{\Pi_a(\Pi^n)}$.*

Theorem 2 implies that once we have found $\bar{\mathbf{w}}^{\text{SMP}}_{\Pi^n}$ we can remove the inactive policies from the set and still guarantee the same worst case performance. Furthermore, we can continue with Algorithm 1 to find the next policy by maximizing $\bar{\mathbf{w}}^{\text{SMP}}_{\Pi^n}$ and guarantee strict improvement via Theorem 1. This is important in applications that have memory constraints, since it allows us to store fewer policies.

## 5 Experiments

We begin with a $10 \times 10$ grid-world environment (Fig. 1(d)), where the agent starts in a random place in the grid (marked in a black color) and gains/loses reward from collecting items (marked with white color). Each item belongs to one of $d - 1$ classes (here with $d = 5$) and is associated with a marker: $8, O, X, Y$. In addition, there is one "no item" feature (marked in gray color). The features are one-hot vectors, *i.e.*, for $i \in [1, d-1]$, $\phi^i(s)$ equals one when item $i$ is in state $s$ and zero otherwise (similarly $\phi^d(s)$ equals one when there is no item in state $s$). The objective of the agent is to pick up the "good" objects and avoid "bad" objects, depending on the weights of the vector $\mathbf{w}$.

In Fig. 1(a) we report the performance of the SMP $\pi^{\text{SMP}}_{\Pi^t}$ w.r.t $\bar{\mathbf{w}}^{\text{SMP}}_{\Pi^t}$ for $d = 5$. At each iteration (x-axis) of Algorithm 1 we train a policy for $5 \cdot 10^5$ steps to maximize $\bar{\mathbf{w}}^{\text{SMP}}_{\Pi^t}$. We then compute the SFs of that policy using additional $5 \cdot 10^5$ steps and evaluate it w.r.t $\bar{\mathbf{w}}^{\text{SMP}}_{\Pi^t}$.

As we can see, the performance of SMP strictly improves as we add more policies to the set (as we stated in Theorem 1). In addition, we compare the performance of SMP with that of GPI, defined on the same sets of policies ($\Pi^t$) that were discovered by Algorithm 1. Since we do not know how to compute $\bar{\mathbf{w}}^{\text{GPI}}_{\Pi^t}$ (the worst case reward for GPI), we evaluate GPI w.r.t $\bar{\mathbf{w}}^{\text{SMP}}_{\Pi^n}$ (the blue line in Fig. 1(a)).

Inspecting Fig. 1(a), we can see that the GPI policy indeed performs better than the SMP as Lemma 2 indicates. We note that the blue line (in Fig. 1(a)) does not correspond to the worst case performance of the GPI policy. Instead, we can get a good approximation for it because we have that: $\bar{\mathbf{w}}^{\text{SMP}}_{\Pi^n} \cdot$

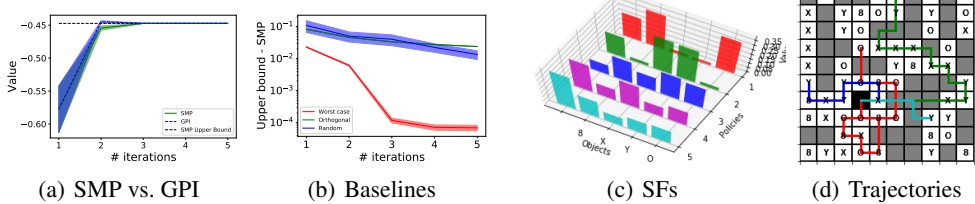

| (a) SMP vs. GPI | (b) Baselines | (c) SFs | (d) Trajectories |

Figure 1: Experimental results in a 2D grid world. Fig. 1(a) presents the performance of the SMP and GPI w.r.t the worst case reward. Fig. 1(b) compares Algorithm 1 with two baselines, where we show the worst case performance, relative to the upper bound, in a logarithmic scale. Fig. 1(c) visualizes the SFs of the policies in the set and Fig. 1(d) presents trajectories that were taken by different policies.

$\psi(\pi^{\text{SMP}}_{\Pi^n}) \leq \bar{\mathbf{w}}^{\text{GPI}}_{\Pi^n} \cdot \psi(\pi^{\text{GPI}}_{\Pi^n}) \leq \bar{\mathbf{w}}^{\text{SMP}}_{\Pi^n} \cdot \psi(\pi^{\text{GPI}}_{\Pi^n})$; i.e., the worst case performance of GPI (in the middle) is guaranteed to be between the green and blue lines in Fig. 1(a). This also implies that the upper bound in Lemma 3 does not apply for the blue line.

We also compare our algorithm to two baselines in Fig. 1(b) (for $d = 10$): (i) Orthogonal - at iteration $t$ we train policy $\pi^t$ to maximize the reward $\mathbf{w} = \mathbf{e}^t$ (a vector of zeroes with a one on the t-th coordinate) such that a matrix with the vectors $\mathbf{w}$ in its columns forms the identity matrix; (ii) Random: at iteration $t$ we train policy $\pi^t$ to maximize reward $\mathbf{w} \sim \tilde{N}(\bar{0}, \bar{1})$, *i.e.*, we sample a vector of dimension $d$ from a Normal Gaussian distribution and normalize it to have a norm of 1. While all the methods improve as we add policies to the set, Algorithm 1 clearly outperforms the baselines.

In Fig. 1(c) and Fig. 1(d) we visualize the policies that were discovered by Algorithm 1. Fig. 1(c) presents the SFs of the discovered policies, where each row (color) corresponds to a different policy and the columns correspond to the different features. We do not enumerate the features from 1 to $d$, but instead we label them with markers that correspond to specific items (the x-axis labels). In Fig. 1(d) we present a trajectory from each policy. We note that both the colors and the markers match between the two figures: the red color corresponds to the same policy in both figures, and the item markers in Fig. 1(d) correspond to the coordinates in the x-axis of Fig. 1(c).

Inspecting the figures we can see that the discovered policies are qualitatively diverse: in Fig. 1(c) we can see that the SFs of the different policies have different weights for different items, and in Fig. 1(d) we can see that the policies visit different states. For example, we can see that the teal policy has a larger weight for the no item feature (Fig. 1(c)) and visits only no-item states (Fig. 1(d)) and that the green policy has higher weights for the 'Y' and 'X' items (Fig. 1(c)) and indeed visits them (Fig. 1(d)).

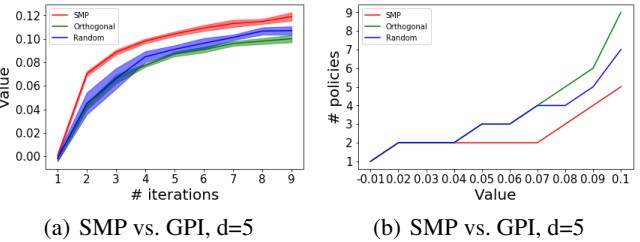

| (a) SMP vs. GPI, d=5 | (b) SMP vs. GPI, d=5 |

Figure 2: Experimental results with regularized $w$.

Finally, in Fig. 2, we compare the performance of our algorithm with that of the baseline methods over a test set of rewards. The only difference is in how we evaluate the algorithms. Specifically, we sampled 500 reward signals from the uniform distribution over the unit ball. Recall that at iteration $t$ each algorithm has a set of policies $\Pi^t$, so we evaluate the SMP defined on this set, $\pi^{\text{SMP}}_{\Pi^t}$, w.r.t each one of the test rewards. Then, for each method, we report the mean value obtained over the test rewards and repeat this procedure for 10 different seeds. Finally, we report the mean and the confidence interval over the seeds. Note that the performance in this experiment will necessarily be better than the in Fig. 1(a) because here we evaluate average performance rather than worst-case performance. Also note that our algorithm was not designed to optimize the performance on this "test set", but to optimize the performance w.r.t the worst case. Therefore it is not necessarily expected to outperform the baselines when measured on this metric.

Inspecting Figure Fig. 2(a) we can see that our algorithm (denoted by SMP) performs better than the two baselines. This is a bit surprising for the reasons mentioned above, and suggests that optimising for the worst case also improves the performance w.r.t the entire distribution (transfer learning result). At first glance, the relative gain in performance might seem small. Therefore, the baselines might seem preferable to some users due to their simplicity. However, recall that the computational cost for computing the worst case reward is small compared to finding the policy the maximizes it, and therefore the relative cost of the added complexity is low.

The last observation suggests that we should care about how many policies are needed by each method to achieve the same value. We present these results in Fig. 2(b). Note that we use exactly the same data as in Fig. 2(a) but present it in a different manner. Inspecting the figure, we can see that the baselines require more policies to achieve the same value. For example, to achieve a value of $0.07$, the SMP required $2$ policies, while the baselines needed $4$; and for a value of $0.1$ the SMP required $4$ policies while the baselines needed $7$ and $9$ respectively.

**DeepMind Control Suite.** Next, we conducted a set of experiments in the DM Control Suite (Tassa et al., 2018). We focused on the setup where the agent is learning from feature observations corresponding to the positions and velocities of the "body" in the task (pixels were only used for visualization). We considered the following six domains: 'Acrobot', 'Cheetah', 'Fish', 'Hopper', 'Pendulum', and 'Walker'. In each of these tasks we do not use the extrinsic reward that is defined by the task, but instead consider rewards that are linear in the observations (of dimensions 6, 17, 21, 15, 3, and 24 respectively). At each iteration of Algorithm 1 we train a policy for $2 \cdot 10^6$ steps using an actor-critic (and specifically STACX (Zahavy et al., 2020d)) to maximize $\bar{\mathbf{w}}_{\Pi^t}^{\text{SMP}}$, add it to the set, and compute a new $\bar{\mathbf{w}}_{\Pi^{t+1}}^{\text{SMP}}$.

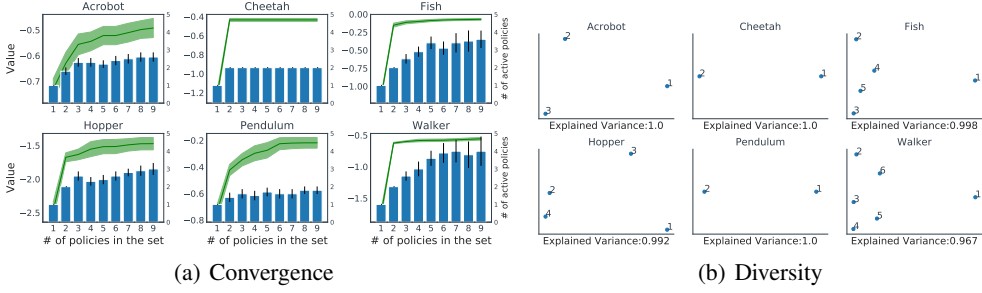

(a) Convergence          (b) Diversity

Figure 3: Experimental results in Deepmind Control Suite.

Fig. 3(a) presents the performance of SMP in each iteration w.r.t $\bar{\mathbf{w}}_{\Pi^t}^{\text{SMP}}$. As we can see, our algorithm is indeed improving in each iteration. In addition, we present the average number of active policies (Definition 6) in each iteration with bars. All the results are averaged over 10 seeds and presented with $95\%$ Gaussian confidence intervals. Fig. 3(b) presents the SFs of the active policies at the end of training (the seed with the maximum number of active policies was selected). We perform PCA dimensionality reduction such that each point in the scatter plot corresponds to the SFs of one of the active policies. We also report the variance explained by PCA: values close to $1$ indicate that the dimensionality reduction has preserved the original variance. Examining the figures we can see that our algorithm is strictly improving (as Theorem 1 predicts) and that the active policies in the set are indeed diverse; we can also see that adding more policies is correlated with improving performance.

Finally, in Fig. 4(a), Fig. 4(b) and Fig. 4(c) we visualize the trajectories of the discovered policies in the Cheetah, Hopper and Walker environments. Although the algorithm was oblivious to the extrinsic reward of the tasks, it was still able to discover different locomotion skills, postures, and even some "yoga poses" (as noted by the label we gave each policy on the left). The other bodies (Acrobot, Pendulum and Fish) have simpler bodies and exhibited simpler movement in various directions and velocities, *e.g.* the Pendulum learned to balance itself up and down. The supplementary material contains **videos** from all the bodies.

## 6 CONCLUSION

We have presented an algorithm that incrementally builds a set of policies to solve a collection of tasks defined as linear combinations of known features. The policies returned by our algorithm can be composed in multiple ways. We have shown that when the composition is a SMP its worst-case

performance on the set of tasks will strictly improve at each iteration of our algorithm. More generally, the performance guarantees we have derived also serve as a lower bound for any composition of policies that qualifies as a SIP. The composition of policies has many applications in RL, such as for example to build hierarchical agents or to tackle a sequence of tasks in a continual learning scenario. Our algorithm provides a simple and principled way to build a diverse set of policies that can be used in these and potentially many other scenarios.

# 7 ACKNOWLEDGEMENTS

We would like to thank Remi Munos and Will Dabney for their comments and feedback on this paper.

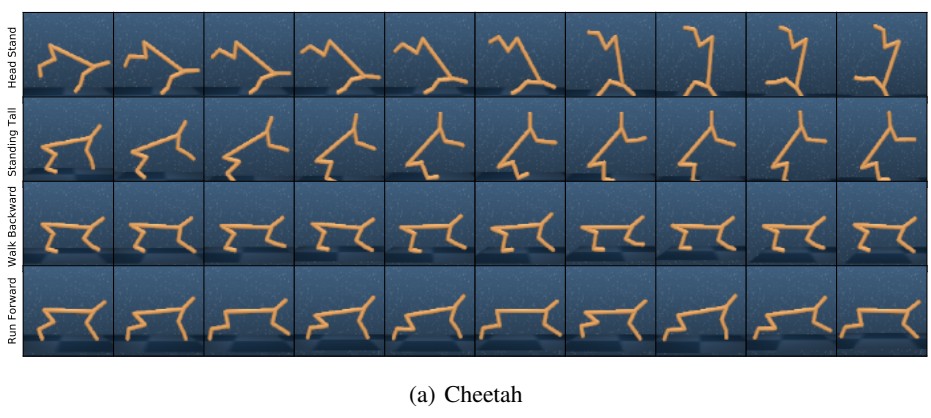

(a) Cheetah

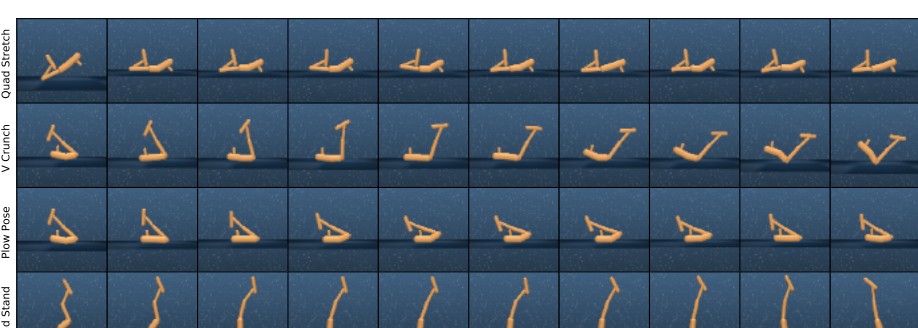

(b) Hopper

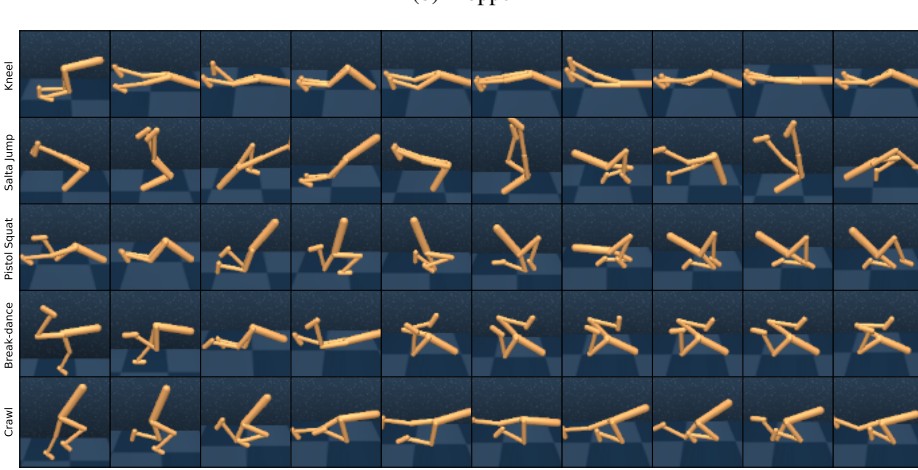

(c) Walker

Figure 4: Experimental results in Deepmind Control Suite.

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

## A    PROOFS

**Lemma 1.** *Let $\pi_{\Pi^n}^{SMP}$ be a SMP defined as in (5) and let $\pi : S \times \mathcal{W} \mapsto \mathbb{P}(A)$ be any generalized policy. Then, given a set of $n$ policies $\Pi^n$, $\pi$ is a SIP if and only if $v_{\Pi^n,\mathbf{w}}^{\pi} \geq v_{\Pi^n,\mathbf{w}}^{SMP}$ for all $\mathbf{w} \in \mathcal{W}$.*

*Proof.* We first show that the fact that $\pi$ is a SIP implies that $v_{\Pi^n,\mathbf{w}}^{\pi} \geq v_{\Pi^n,\mathbf{w}}^{SMP}$ for all $\mathbf{w}$. For any $\mathbf{w} \in \mathcal{W}$, we have

$$
\begin{aligned}
v_{\Pi^n,\mathbf{w}}^{\pi} &\geq v_{\mathbf{w}}^i \text{ for all } \pi^i \in \Pi^n \qquad\qquad\qquad\text{(SIP as in Definition 2)}\\
&\geq \max_{i \in [1,\ldots,n]} v_{\mathbf{w}}^i \\
&= v_{\Pi^n,\mathbf{w}}^{SMP}.
\end{aligned}
$$

We now show the converse:

$$
\begin{aligned}
v_{\Pi^n,\mathbf{w}}^{\pi} &\geq v_{\Pi^n,\mathbf{w}}^{SMP} \\
&= \max_{i \in [1,\ldots,n]} v_{\mathbf{w}}^i \qquad\qquad\qquad\text{(SMP as in Definition 3)}\\
&\geq v_{\mathbf{w}}^i \text{ for all } \pi^i \in \Pi^n.
\end{aligned}
$$

∎

**Lemma 2.** *For any reward $\mathbf{w} \in \mathcal{W}$ and any set of policies $\Pi^n$, we have that $v_{\Pi^n,\mathbf{w}}^{GPI} \geq v_{\Pi^n,\mathbf{w}}^{SMP}$.*

*Proof.* We know from previous results in the literature Barreto et al. (2017) that $Q^{\text{GPI}}(\Pi^n)(s,a) \geq Q^{\pi}(s,a)$ for all $(s,a) \in \mathcal{S} \times \mathcal{A}$ and any $\pi \in \Pi^n$.

Thus, we have that $\forall s \in S$:

$$
\begin{aligned}
v_{\Pi^n,\mathbf{w}}^{\text{GPI}}(s) &= Q_{\Pi^n,\mathbf{w}}^{\text{GPI}}(s, \pi^{\text{GPI}}(s)) \\
&\geq \max_{\pi \in \Pi^n, a \in \mathcal{A}} Q_{\mathbf{w}}^{\pi}(s,a) \\
&\geq \max_{\pi \in \Pi^n} \mathbb{E}_{a \sim \pi}[Q_{\mathbf{w}}^{\pi}(s,a)] \\
&= \max_{\pi \in \Pi^n} v_{\mathbf{w}}^{\pi}(s) \\
&= v_{\Pi^n,\mathbf{w}}^{SMP}(s),
\end{aligned}
$$

where the second inequality is due to Jensen's inequality.

Therefore:

$$
\begin{aligned}
v_{\Pi^n,\mathbf{w}}^{\text{GPI}}(s) &\geq v_{\Pi^n,\mathbf{w}}^{SMP}(s) \\
\mathbb{E}_D[v_{\Pi^n,\mathbf{w}}^{\text{GPI}}(s)] &\geq \mathbb{E}_D[v_{\Pi^n,\mathbf{w}}^{SMP}(s)] \\
v_{\Pi^n,\mathbf{w}}^{\text{GPI}} &\geq v_{\Pi^n,\mathbf{w}}^{SMP}
\end{aligned}
$$

∎

**Lemma 3** (Impossibility result). *For the special case where the SFs associated with any policy are in the simplex, the value of the SMP w.r.t the worst case reward for any set of policies is less than or equal to $-1/\sqrt{d}$. In addition, there exists an MDP where this upper bound is attainable.*

*Proof.* For the **impossibility** result. we have that

$$\max_{\Pi^n \subseteq \Pi} \min_{w \in B_2} v_{\Pi^n, \mathbf{w}}^{\text{SMP}} = \min_{w \in B_2} \max_{\pi \in \Pi} v_w^{\pi} \tag{9}$$

$$= \min_{w \in B_2} \max_{\pi \in \Pi} \psi(\pi) \cdot w$$

$$\leq \min_{w \in B_2} \max_{\psi \in \Delta^{d-1}} \psi(\pi) \cdot w \tag{10}$$

$$= \min_{w \in B_2} \max_{i} w_i \tag{11}$$

$$= -\frac{1}{\sqrt{d}}. \tag{12}$$

The equality in Eq. (9) follows from the fact that $\Pi$ is set of all possible policies and therefore the largest possible subset (the maximizer of the first maximization). In that case the second maximization (by the SMP) is equivalent to selecting the optimal policy in the MDP. Notice that the order of maximization-minimization here is in the reversed when compared to AL, i.e., for each reward the SMP chooses the best policy in the MDP, while in AL the reward is chosen to be the worst possible w.r.t any policy. The inequality in Eq. (10) follows from the fact that we increase the size of the optimization set in the inner loop, and the equality in Eq. (12) follows from the fact that a maximizer in the inner loop puts maximal distribution on the largest component of $w$.

**Feasibility.** To show the feasibility of the upper bound in the previous impossibility result we give an example of an MDP in which a set of $d$ policies achieves the upper bound. The $d$ policies are chosen such that their stationary distributions form an orthogonal basis.

$$\min_{w \in B_2} v_{\Pi^n, \mathbf{w}}^{\text{SMP}} = \min_{w \in B_2} \max_{\psi \in \{\psi^1, \ldots, \psi^d\}} w \cdot \psi = \min_{w \in B_2} \max_{\psi \in \Delta^{d-1}} w \cdot \psi = -\frac{1}{\sqrt{d}}, \tag{13}$$

which follows from the fact that the maximization over the simplex is equivalent to a maximization over pure strategies. ∎

**Lemma 4** (Reformulation of the worst-case reward for an SMP). *Let $\{\psi^i\}_{i=1}^n$ be $n$ successor feature vectors. Let $\mathbf{w}^*$ be the adversarial reward, w.r.t the SMP defined given these successor features. That is, $\mathbf{w}^*$ is the solution for*

$$\arg\min_w \quad \max_{i \in [1, \ldots, n]} \{w \cdot \psi^1, \ldots, w \cdot \psi^n\}$$

$$s.t. \quad ||w||^2 - 1 \leq 0 \tag{14}$$

*Let $w_i^*$ be the solution to the following problem for $i \in [1, \ldots, n]$:*

$$\arg\min_w \quad w \cdot \psi^i$$

$$s.t. \quad ||w||^2 - 1 \leq 0$$

$$w \cdot (\psi^j - \psi^i) \leq 0 \tag{15}$$

*Then, $\mathbf{w}^* = \arg\min_i w_i^*$.*

*Proof.* For any solution $w^*$ to Eq. (8) there is some policy $i$ in the set that is one of its maximizers. Since it is the maximizer w.r.t $w^*$, its value w.r.t $w^*$ is bigger or equal to that of any other policy in the set. Since we are checking the solution among all $i \in [1, \ldots, n]$, one of them must be the solution. ∎

**Theorem 2** (Sufficiency of Active policies). *For any set of policies $\Pi^n$, $\pi_{\Pi_a(\Pi^n)}^{SMP}$ achieves the same value w.r.t the worst case reward as $\pi_{\Pi^n}^{SMP}$, that is, $\bar{v}_{\Pi^n}^{SMP} = \bar{v}_{\Pi_a(\Pi^n)}^{SMP}$.*

*Proof.* Let $\Pi^n = \{\pi_i\}_{i=1}^n$. Denote by $J$ a subset of the indices $[1, \ldots, n]$ that corresponds to the indices of the active policies such that $\Pi_a(\Pi^n) = \{\pi_j\}_{j \in J}$. We can rewrite problem Eq. (14) as follows:

$$\begin{aligned} \text{minimize} \quad & \gamma \\ s.t. \quad & \gamma \geq w \cdot \psi^i \quad i = 1, \ldots, n \\ & \|w\|^2 \leq 1. \end{aligned} \tag{16}$$

Let $(\gamma^\star, w^\star)$ be any optimal points. The set of inactive policies $i \notin \mathcal{J}$ satisfy $\gamma^\star > w^\star \cdot \psi^i$. Since these constraints are not binding we can drop them from the formulation and maintain the same optimal objective value, *i.e.*,

$$
\begin{aligned}
\text{minimize} \quad & \gamma \\
\text{s.t.} \quad & \gamma \geq w \cdot \psi^j \quad j \in \mathcal{J} \\
& \|w\|^2 \leq 1,
\end{aligned}
\tag{17}
$$

has the same optimal objective value, $\bar{v}^{\mathrm{SMP}}_{\Pi^n}$, as the full problem. This in turn can be rewritten

$$
\begin{aligned}
\text{minimize} \quad & \max_{j \in \mathcal{J}} w \cdot \psi^j \\
\text{s.t.} \quad & \|w\|^2 \leq 1,
\end{aligned}
\tag{18}
$$

with optimal value $\bar{v}^{\mathrm{SMP}}_{\Pi_a(\Pi^n)}$, which is therefore equal to $\bar{v}^{\mathrm{SMP}}_{\Pi^n}$.

$\blacksquare$

**Lemma 5** ($\kappa$ is binding).

*Proof.* Denote by $\dot{w}$ a possible solution where the constraint $\|\dot{w}\|^2 \leq 1$ is not binding, i.e., ($\|\dot{w}\|^2 < 1, \dot{\kappa} = 0$). In addition, denote the primal objective for $\dot{w}$ by $\dot{v} = \max_{i \in [1,...,n]} \{ \dot{w} \cdot \psi^i \}$. To prove the lemma, we are going to inspect two cases: (i) $\dot{v} \geq 0$ and (ii) $\dot{v} < 0$. For each of these two cases we will show that there exists another feasible solution $\tilde{w}$ that achieves a lower value $\tilde{w}$ for the primal objective ($\tilde{w} < \dot{v}$), and therefore $\dot{w}$ is not the minimizer.

For the first case $\dot{v} \geq 0$, consider the vector

$$
\tilde{w} = (-1, -1, \ldots, -1)/\sqrt{d}.
$$

$\tilde{w}$ is a feasible solution to the problem, since $\|\tilde{w}\|^2 = 1$. Since all the SFs have positive coordinates, we have that if they are not all exactly $0$, then the primal objective evaluated at $\tilde{w}$ is stictly negative: $\max_{i \in [1,...,n]} \{ \tilde{w} \cdot \psi^1, \ldots, \tilde{w} \cdot \psi^n \} < 0$.

We now consider the second case of $\dot{v} < 0$. Notice that multiplying $\dot{w}$ by a positive constant $c$ would not change the maximizer, i.e., $\arg \max_{i \in [1,...,n]} \{ c\dot{w} \cdot \psi^i \} = \arg \max_{i \in [1,...,n]} \{ \dot{w} \cdot \psi^i \}$. Since $\dot{v} < 0$, it means that $\dot{w}/\|\dot{w}\|$ ($c = 1/\|\dot{w}\|$) is a feasible solution and a better minimizer than $\dot{w}$. Therefore $\dot{w}$ is not the minimizer.

We conclude that the constraint $\kappa$ is always binding, i.e. $\|w\|^2 = 1, \kappa > 0$. $\blacksquare$

**Theorem 1** (Strict improvement). *Let $\Pi^1, \ldots, \Pi^t$ be the sets of policies constructed by Algorithm 1. We have that the worst-case performance of the SMP induced by these set is strictly improving in each iteration, that is: $\bar{v}^{SMP}_{\Pi^{t+1}} > \bar{v}^{SMP}_{\Pi^t}$. Furthermore, when the algorithm stops, there does not exist a single policy $\pi^{t+1}$ such that adding it to $\Pi^t$ will result in improvement: $\nexists \pi^{t+1} \in \Pi$ s.t. $\bar{v}^{SMP}_{\Pi^t + \{\pi\}} > \bar{v}^{SMP}_{\Pi^t}$.*

*Proof.* We have that

$$
v^{\mathrm{SMP}}_{\Pi^t} = \min_{\mathbf{w} \in \mathbb{B}_2} \max_{\boldsymbol{\psi} \in \Psi^t} \boldsymbol{\psi} \cdot \mathbf{w} \leq \max_{\boldsymbol{\psi} \in \Psi^t} \boldsymbol{\psi} \cdot \bar{\mathbf{w}}^{\mathrm{SMP}}_{\Pi^{t+1}} \leq \max_{\boldsymbol{\psi} \in \Psi^{t+1}} \boldsymbol{\psi} \cdot \bar{\mathbf{w}}^{\mathrm{SMP}}_{\Pi^{t+1}} = v^{\mathrm{SMP}}_{\Pi^{t+1}}.
\tag{19}
$$

The first inequality is true because we replace the minimization over $\mathbf{w}$ with $\bar{\mathbf{w}}^{\mathrm{SMP}}_{\Pi^{t+1}}$, and the second inequality is true because we add a new policy to the set. Thus, we will focus on showing that the first inequality is strict. We do it in two steps. In the first step, we will show that the problem $\min_{\mathbf{w} \in \mathbb{B}_2} \max_{\boldsymbol{\psi} \in \Psi^t} \boldsymbol{\psi} \cdot \mathbf{w}$ has a unique solution $\mathbf{w}^\star_t$. Thus, for the first inequality to hold with equality it must be that $\bar{\mathbf{w}}^{\mathrm{SMP}}_{\Pi^{t+1}} = \bar{\mathbf{w}}^{\mathrm{SMP}}_{\Pi^t}$. However, we know that, since the algorithm did not stop, $\boldsymbol{\psi}^{t+1} \cdot \bar{\mathbf{w}}^{\mathrm{SMP}}_{\Pi^t} > v^{\mathrm{SMP}}_{\Pi^t}$, hence a contradiction.

We will now show that $\min_{\mathbf{w} \in \mathbb{B}_2} \max_{\boldsymbol{\psi} \in \Psi^t} \boldsymbol{\psi} \cdot \mathbf{w}$ has a unique solution. Before we begin, we refer the reader to Lemma 4 and Theorem 2 where we reformulate the problem to a form that is simpler to analyze. We begin by looking at the partial Lagrangian of Eq. (17):

$$
L(w, \gamma, \kappa, \lambda) = \gamma + \sum_{j \in J} \lambda_j (\psi^j \cdot w - \gamma) + \kappa (\|w\|^2 - 1).
$$

The variable $\kappa \geq 0$ is associated with the constraint $\|w\|^2 \leq 1$. Denote by $(\lambda^\star, \kappa^\star)$ any optimal dual variables and note that by complementary slackness we know that either $\kappa^\star > 0$ and $\|w\|_2 = 1$ or $\kappa^\star = 0$ and $\|w\|_2 < 1$. Lemma 5 above, guarantees that the constraint is in fact binding – only solutions with $\kappa^\star > 0$ and $\|w\|_2 = 1$ are possible solutions. Notice that this is correct due to the fact that the SFs have positive coordinates and not all of them are 0 (as in our problem formulation).

Consequently we focus on the case where $\kappa^\star > 0$ under which the Lagrangian is strongly convex in $w$, and therefore the problem
$$\min_{w,\gamma} L(w, \gamma, \lambda^\star, \kappa^\star)$$
has a unique solution. Every optimizer of the original problem must also minimize the Lagrangian evaluated at an optimal dual value, and since this minimizer is unique, it implies that the minimizer of the original problem is unique (Boyd et al., 2004, Sect. 5.5.5).

For the second part of the proof, notice that if the new policy $\pi^{t+1}$ does not achieve better reward w.r.t $v_{\Pi^t}^{\text{SMP}}$ than the policies in $\Pi^t$ then we have that:
$$v_{\Pi^{t+1}}^{\text{SMP}} = \min_{w \in \mathbb{B}_2} \max_{\pi \in \Pi^{t+1}} \psi(\pi) \cdot w \leq \max_{\pi \in \Pi^{t+1}} \psi(\pi) \cdot \bar{\mathbf{w}}_{\Pi^t}^{\text{SMP}} = \max_{\pi \in \Pi^t} \psi(\pi) \cdot \bar{\mathbf{w}}_{\Pi^t}^{\text{SMP}} = v_{\Pi^t}^{\text{SMP}};$$

thus, it is necessary that the policy $\pi^{t+1}$ will achieve better reward w.r.t $v_{\Pi^t}^{\text{SMP}}$ to guarantee strict improvement.

$\blacksquare$

## B    AL

In AL there is no reward signal, and the goal is to observe and mimic an expert. The literature on AL is quite vast and dates back to the work of (Abbeel & Ng, 2004), who proposed a novel framework for AL. In this setting, an expert demonstrates a set of trajectories that are used to estimate the SFs of its policy $\pi_E$, denoted by $\psi^E$. The goal is to find a policy $\pi$, whose SFs are close to this estimate, and hence will have a similar return with respect to any weight vector $\mathbf{w}$, given by

$$\arg \max_\pi \min_{\mathbf{w} \in B_2} \mathbf{w} \cdot \left(\psi^\pi - \psi^E\right) = \arg \max_\pi -||\psi^\pi - \psi^E||$$
$$= \arg \min_\pi ||\psi^\pi - \psi^E||. \tag{20}$$

The projection algorithm (Abbeel & Ng, 2004) solves this problem in the following manner. The algorithm starts with an arbitrary policy $\pi_0$ and computes its feature expectation $\psi^0$. At step $t$, the reward function is defined using weight vector $w_t = \psi^E - \bar{\psi}^{t-1}$ and the algorithm finds a policy $\pi_t$ that maximizes it, where $\bar{\psi}^t$ is a convex combination of SFs of previous (deterministic) policies $\bar{\psi}_t = \sum_{j=1}^t \alpha_j \psi^j$. In order to get that $\|\bar{\psi}_T - \psi^E\| \leq \epsilon$, the authors show that it suffices to run the algorithm for $T = O(\frac{k}{(1-\gamma)^2\epsilon^2} \log(\frac{k}{(1-\gamma)\epsilon}))$ iterations.

Recently, it was shown that this algorithm can be viewed as a Frank-Wolfe method, also known as the Conditional Gradient (CG) algorithm (Zahavy et al., 2020a). The idea is that solving Eq. (20) can be seen as a constrained convex optimization problem, where the optimization variable is the SFs, the objective is convex, and the SFs are constrained to be in the SFs polytope $\mathcal{K}$, given as the following convex set:

**Definition 7** (The SFs polytope). $\mathcal{K} = \left\{x : \sum_{i=1}^{k+1} a_i \psi^i, a_i \geq 0, \sum_{i=1}^{k+1} a_i = 1, \pi^i \in \Pi\right\}.$

In general, convex optimization problems can be solved via the more familiar projected gradient descent algorithm. This algorithm takes a step in the reverse gradient direction $z_{t+1} = x_t + \alpha_t \nabla_h(x_t)$, and then projects $z_{t+1}$ back into $\mathcal{K}$ to obtain $x_{t+1}$. However, in some cases, computing this projection may be computationally hard. In our case, projecting into $\mathcal{K}$ is challenging since it has $|A|^{|S|}$ vertices (feature expectations of deterministic policies). Thus, computing the projection explicitly and then finding $\pi$ whose feature expectations are close to this projection, is computationally prohibitive.

The CG algorithm (Frank & Wolfe, 1956) (Algorithm 2) avoids this projection by finding a point $y_t \in \mathcal{K}$ that has the largest correlation with the negative gradient. In AL, this step is equivalent to finding a policy whose SFs has the maximal inner product with the current gradient, i.e., solve an MDP whose reward vector $\mathbf{w}$ is the negative gradient. This is a standard RL (planning) problem and

can be solved efficiently, for example, with policy iteration. We also know that there exists at least one optimal deterministic policy for it and that PI will return a solution that is a deterministic policy (Puterman, 1984).

---

**Algorithm 2** The CG method Frank & Wolfe (1956)

---

1: Input: a convex set $\mathcal{K}$, a convex function $h$, learning rate schedule $\alpha_t$.
2: Initiation: let $x_0 \in \mathcal{K}$
3: **for** $t = 1, \ldots, T$ **do**
4: $\quad y_t = \arg\max_{y \in \mathcal{K}} -\nabla_h(x_{t-1}) \cdot y$
5: $\quad x_t = (1 - \alpha_t)x_{t-1} + \alpha_t y_t$
6: **end for**

---

For smooth functions, CG requires $O(1/t^2)$ iterations to find an $\epsilon-$optimal solution to Eq. (20). This gives a logarithmic improvement on the result of (Abbeel & Ng, 2004). In addition, it was shown in (Zahavy et al., 2020a) that since the optimization objective is strongly convex, and the constraint set is a polytope, it is possible to use a variant of the CG algorithm, known as Away steps conditional gradient (ASCG) (Wolfe, 1970). ASCG attains a linear rate of convergence when the set is a polytope (Guélat & Marcotte, 1986; Garber & Hazan, 2013; Jaggi, 2013), i.e., it converges after $O(\log(1/\epsilon))$ iterations. See (Zahavy et al., 2020a) for the exact constants and analysis.

There are some interesting relations between our problem and AL with "no expert", that is, solving

$$\arg\min_{\pi} ||\psi^{\pi}|| \tag{21}$$

In terms of optimization, this problem is equivalent to Eq. (20), and the same algorithms can be used to solve them.

Both AL with "no expert" and our algorithm can be used to solve the same goal: achieve good performance w.r.t the worst case reward. However, AL is concerned with finding a single policy, while our algorithm is explicitly designed to find a set of policies. There is no direct connection between the policies that are discovered from following these two processes. This is because the intrinsic rewards that are maximised by each algorithm are essentially different. Another way to think about this is that since the policy that is returned by AL is a mixed policy, its goal is to return a set of policies that are similar to the expert, but not diverse from one another. From a geometric perspective, the policies returned by AL are the nodes of the face in the polytope that is closest to the demonstrated SFs. Even more concretely, if the SFs of the expert are given exactly (instead of being approximated from trajectories), then the AL algorithm would return a single vertex (policy). Finally, while a mixed policy can be viewed as a composition of policies, it is not a SIP. Therefore, it does not encourage diversity in the set.

## C  REGULARIZING W

In this section we experimented with constraining the set of rewards to include only vectors $w$ whose mean is zero. Since we are using CVXPY (Diamond & Boyd, 2016) to optimize for $w$ (Eq. (8)), this requires adding a simple constraint $\sum_{i=1}^{d} w_i = 0$ to the minimization problem. Note that constraining the mean to be zero does not change the overall problem qualitatively, but it does potentially increase the difference in the relative magnitude of the elements in $w$. Since it makes the resulting $w$'s have more zero elements, i.e., it makes the $w$'s more sparse, it can also be viewed as a method to regularize the worst case reward. Adding this constraint increased the number of $w$'s (and corresponding policies) that made a difference to the optimal value (Definition 5). To see this, note that the green curve in Fig. 5(a) converges to the optimal value in 2 iterations while the the green curve in Fig. 1(a)) does so in 3 iterations. As a result, the policies that were discovered by the algorithm are more diverse. To see this observe that the SFs in Fig. 5(b) are more focused on specific items than the SFs in Fig. 1(c). In Fig. 5(c) and Fig. 5(d) we verified that this increased diversity continues to be the case when we increase the feature dimension $d$.

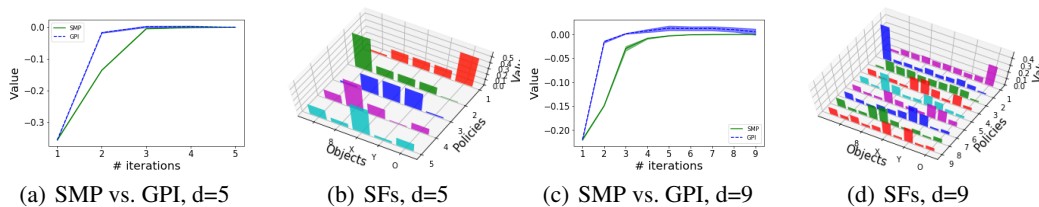

(a) SMP vs. GPI, d=5  (b) SFs, d=5  (c) SMP vs. GPI, d=9  (d) SFs, d=9

Figure 5: Experimental results with regularized $w$.

