# OpenReview forum: "Discovering a set of policies for the worst case reward"
_ICLR.cc/2021/Conference — ICLR 2021 Spotlight_

### Official Review · AnonReviewer1 · 2020-10-26
**Overall interesting idea. Need some clarification.**

**Rating:** 6
**Confidence:** 4

**Review:**

This was a well-written, and interesting paper to read!

I went over the paper many times, and I am still failing to see the use case for such an approach. I have some questions and comments that need some clarification for me to properly evaluate the submission. Please, take the time to answer the following, so my review can better reflect the paper.

1 - The theory developed in the paper relies on reward functions that can be represented as linear combinations of the features of the MDP. This seems to be restrictive, and intuitively, this would be the exception rather than the rule.
What class of problem could be modeled under this restriction?  In many problems, there is no linear reward function that would allow an agent to achieve the desired behavior, so these techniques would not be helpful. What are some practical setting where this approach would be beneficial?

2 - Lemma 3... this statement is putting an upper-bound on the worst case performance, but since the paper focuses on improvement of worst case performance, it would be beneficial to have a lower bound, but an upper bound doesn't seem too useful.   Essentially, this lemma is saying "I can guarantee that the worst-case won't be better than this upper bound, and that for some MDP with linear reward function this upper bound is attainable." The problem is that we don't know what that MDP is, how likely it is that we would find it, and this lemma allow for the worst case performance to be arbitrarily bad.
I don't think this lemma, as is, is particularly useful.

3 - On the experimental section, I think there's a baseline that should be included that's missing. What if we have 1 policy and add a task descriptor or extra features to the features vector that corresponds to the type of task? How would the performance empirically compare?

4 - In the learning curves for fig 1.a or 2.a, what does "value" (y-axis) represent? If it the return of the agent after training? If so, is it using the extrinsic reward or the transformed linear reward described in line 5 of "DeepMind Control Suite"?

5 - Based on equation 4, for definition 2 of SIP. There is always a trivial set improving policy, right? That would correspond to picking the policy for max(v^i_w).

---

> ### Author Response · Authors · 2020-11-13
> **Response to Reviewer 1**
>
> We would like to thank the reviewer for their feedback. Our response is split to two parts.
>
> Question: “what can be a use case of such an approach?” We believe that we presented an interesting learning framework in this work: in a world without an explicit reward, can an agent define goals to itself and discover interesting behaviors by doing so? Our experiments in DM control suggest that diverse and interesting locomotion skills (such as salta jumping) can emerge from following this process which is an interesting scientific observation. For a more concrete use case, imagine that a robot can teach itself how to move and locomote in an unsupervised discovery phase, later to use these skills when instructed to do more complicated tasks. Since the skills that the robot discovers prepare it for the worst case, some of them are likely to be useful in the future from a robustness perspective, that is, no matter what the robot discovered, we are guaranteed that the robot won’t do too bad when faced with a new task. Thus, the learned skills can also be used to initialize the robot’s behaviour, to be followed by a learning phase. Another interesting contribution of this work is the connection between robustness and diversity in RL: we have demonstrated that by optimizing for the worst case, a diverse set of skills emerges.
>
> Question 1 (the linearity assumption):
>
> It is indeed true that for a *fixed* set of features the linearity assumption is restricting. However, it is this assumption that allows us to develop theory and thus provide theoretical guarantees for the proposed algorithm. Note that the guarantees we provide are in fact applicable in practice, as we illustrate in our experiments. For example, the experiments in the tabular MDP follow the theoretical framework exactly and achieve the upper bound that we developed (see more on that in our answer to point 2 below). The experiments in DM control are also very close to the theory, the only deviation is that we use DRL to optimize the policy for a given reward. We note that this agreement between theory and practice is in fact one of the strengths of our work, since more often than not there is a gap between the two.
>
> All that said, we point out that in the most general scenario we are free to define the features themselves. Note that, although the rewards are linear in the features, the features themselves can be arbitrary nonlinear functions of state-action pairs. This means that when we are able to define the features the linear assumption is in fact not so strong: for example, in [1], the authors discuss how in the tabular case this is not a restriction at all, and how in continuous state spaces we can find features that approximate any reward function with a certain accuracy.
>
> As a somewhat counterintuitive observation, we note that in many problems it is in fact easy to handcraft simple features that generate useful behaviour. This is illustrated in our experiments with DM control, in which using the standard features provided in the suite we were able to generate rich behaviour. For example, we were able to teach the walker to do salta jumps (see the video in the supplementary material) by simply using linear combinations of the standard features.
>
> Thirdly, our experiments suggest that the assumption is not too restricting in interesting problems. For example, we were able to teach the walker to do salta jumps (see the video in the supplementary material) under this assumption.
>
> Lastly, we believe that it should be easy to generalize our approach to a more general setup where the reward is represented as a nonlinear (perhaps a DNN) of the features. In this case, the minimization over w will not be  convex but will still be possible via the same techniques (SGD). This kind of algorithm will resemble GAIL (with a GAN) and we believe that it is an exciting direction of research for future work.
>
>
> [1] Barreto, A., Hou, S., Borsa, D., Silver, D., & Precup, D. "Fast reinforcement learning with generalized policy updates." Proceedings of the National Academy of Sciences (2020).
> [2] Abbeel, Pieter, and Andrew Y. Ng. "Apprenticeship learning via inverse reinforcement learning." Proceedings of the twenty-first international conference on Machine learning. 2004.
> [3] Zahavy, Tom, Alon Cohen, Haim Kaplan, and Yishay Mansour. "Apprenticeship Learning via Frank-Wolfe." AAAI (2020).

---

> > ### Author Response · Authors · 2020-11-13
> > **Response to Reviewer 1, cont.**
> >
> > Question 2: lemma 3.
> > Given a set of policies, we have a lower bound on the worst case performance of the SMP. This is equivalent to computing the worst case reward w.r.t the current set and measuring the performance of the SMP. We do not, however, have a lower bound on what that value would be given that we run our algorithm for n iterations. That would indeed be a great contribution; we will mention it in the paper as a promising direction for future research.
> >
> > That said, we want to emphasize that Lemma 3 is useful, as it provides a clear criterion for the convergence of our algorithm: whenever the upper bound provided in the lemma is achieved, we can stop adding more policies. We would like to point out that this in fact happens in practice, as we illustrate in our experiments (Figure 1a). This is surprising since often upper bounds are not attainable in practice, as the reviewer implied, which makes Lemma 3 even more relevant.
> >
> > Question 3 (experiments).
> > This is an interesting suggestion, and we are working on performing this experiment, although we are not sure if we can make it on time. Our intuition is that that sort of a baseline will be less practical to use as it typically requires many iterations until it will be able to generalize to new tasks (e.g. UVFA [4]) while our algorithm performs well after a few iterations.
> >
> > Question 4, value in figures.
> > The values shown in Figures 1a and 2a correspond to the SMP’s value, given in Definition 5. The way we compute it as follows. For each policy computed by our algorithm, we estimate the associated successor features (SFs) using Monte Carlo estimation: that is, we fix the policy and run at multiple times to estimate the SFs. We run it enough times to guarantee that the estimate is accurate (see, for example, theorem 2 in [2] or lemma 5 in [3],  for a concentration bound on the approximation error for a given number of samples). Now, given a set o n policies pi (and their associated SFs), we compute the worst possible w, which we call w*, (Equation 8). We then compute the inner product between the n SFs and w* and pick the maximum of these n values (Definition 5). This is the value shown in the figures, which represent the worst possible performance of the set of n policies across all possible tasks.
> >
> > Question 5.
> > Yes, you are correct: this SIP is exactly the SMP (Definition 3).

---

> > > ### Comment · AnonReviewer1 · 2020-11-13
> > > **Response to authors**
> > >
> > > Thank you for taking the time to address the comments, I really appreciate the effort.
> > > Below are my comments to the response.
> > >
> > > - What is a use-case for this approach? So, you give an example of a robot learning locomotion in an unsupervised manner, and argue that this would help the agent prepare for worst-case in subsequent tasks. I'm trying to picture what this would look like in practice. Is what you are suggesting that by simply learning locomotion, the agent might not have covered scenarios that would encounter in follow up tasks, so worst-case performance in the locomotion skill could be poor enough that the new tasks becomes unlearnable?
> > >
> > >  - Thanks for the clarification in the linear features. This makes sense.
> > >
> > > -  Thanks for the clarification in figure 4. It would also be useful to see how the actual performance of the agent compares in traditional methods. The fact that the worst-case performance is better  than for other methods would not be very useful, if the best-case performance is not good enough to complete the task. Even if for a few domains/task, I think this figure must be there.
> > > In other words, by ensuring that our worst-case performance is not too bad...what are we losing from the best case performance. Depending on the scenario, it might be acceptable or it might not.
> > > If time permits, please include such results.
> > >
> > > - On Lemma 3...thanks for the clarification. I see now how that would be useful.
> > >
> > > - I understand the time constraint for providing the one baseline I suggested. If you can include that, it will be very appreciated, but if you can't I won't hold it against you :)
> > > If you can only add one of the suggestions I have, please make it the comparison I suggested to add for figure 4.

---

> > > > ### Comment · AnonReviewer1 · 2020-11-13
> > > > **Response to authors #2**
> > > >
> > > > Thank you for addressing the concerns in the above comments.
> > > > It's definitely interesting the fact that the performance seems to be a bit better than the baseline, to be honest, I would have expected it to not perform as well.
> > > >
> > > > Just to clarify, these are the same tasks on the grid-world from Section 5, correct?
> > > >
> > > >
> > > > I'm adjusting my score based on your responses.

---

> > > > > ### Author Response · Authors · 2020-11-13
> > > > > **Response to reviewer #2**
> > > > >
> > > > > We would like to thank the reviewer for replying to our response and for increasing their score. We appreciate your feedback and believe that it helped us to improve the paper.
> > > > >
> > > > > Regarding the clarification question: indeed, these are the tasks in the grid world from Section 5.

---

### Official Review · AnonReviewer2 · 2020-10-26
**Official Blind Review #3**

**Rating:** 7
**Confidence:** 4

**Review:**

Given a rewardless environment MDP, the authors want to find a set of policies for the worst case reward function. Their process involves two steps: first to select the right set of policies and second to combine them to generate a new policy. The policy selection is made with the only goal to maximize the expected return of highest achieving policy of the set in the worst-case reward function (Equation (7)).

Unfortunately the submission suffers from several serious weaknesses:
- although the reward-set setting is quite general: within a ball in the feature space, it includes very unnatural cases that make the problem artificially too complex, in my opinion. Indeed, usually, one should consider that the worst case reward function r_min in a reward function family R is the one that is minimal in every state-action-state : r_min(s,a,s') \eqdef \inf_{r\in R} r(s,a,s'). In this case, the solution to equation (7) is straightforwardly the single policy that optimizes the return on r_min. Please discuss more the interest of considering r=\psi.w with w in a ball (it implies that if some feature takes sometimes positive and sometimes negative values, there is not clear w_min, and therefore no clear r_min).
- besides ensuring that SMP performance is at least achieved, could the authors elaborate a bit more on why optimize the policy set according to SMP?
- the authors never theoretically consider combining the policy, apart for stating that a good combination of policy should achieve higher performance than the best of the policy set. For clarity, I would recommend to either stick to the simplest setting of choosing the best policy given a reward function, or to consider policy election that take into account the way the policies are going to be used/combined.
- the formalization is messy and sometimes unnecessarily confusing. Please see the series of comments below:
- Definition 3 and Eq. 5: the argmax returns an index, not a policy. (minor)
- v is not a value, it's a policy performance. It has been very confusing to me, as it led me to think for too long that Lemma 1 was false: choosing the policy that maximizes the value in each state is a form policy improvement that may lead to policies that are stricty better than the best of the policies. Also, I would not use the notation v, that is usually the value-function: a function of the station and not the policy performance like here, i.e. the expectation of the value-function over the initial states distribution. (easy to fix)
- Definition 4: the argmax returns an action not a policy. (minor)
- Equation 7: \Pi lives in? Also instead of max_{\psi\in...} \psi.w, I would use max_{i\in [n]} \psi^i.w. (minor)
- Lemma 3: what is d? (minor)
- I am still not understanding Definition 6 and Theorem 2. How do we know that the worst-case reward is unique? If we keep only the policies that achieve max performance on \overline{w}, then we probably only keep one? How do we ensure that there is not another w that makes this policy (or set of policies) to fail?

For all these reasons, I recommend to reject the submission.

---

> ### Author Response · Authors · 2020-11-13
> **Response to Reviewer 2**
>
> We would like to thank the reviewer for their constructive comments, we are sure that our changes to the manuscript following their suggestion have improved the quality of this work.
>
> Please note that we made substantial changes to the paper following your suggestions. This includes addressing the $\arg\max$ in Definitions 3&4, and in Eq 5, and switching the notation in Eq 7. Regarding the question “where does $\Pi$ live in?”: this is a good point, since the original submission confusingly used the same notation $\Pi$ for the subset of policies and the set of all the policies in the MDP. We corrected that in the revised draft, such that it's clear that we optimize over a subset $\Pi^n$ of $\Pi$ where $\Pi$ is the set of all the policies.
>
> Scalar value functions. Thank you for the comment, we have clarified the notation regarding the value in Equation 3 to clarify that it is the expected value under the initial state distribution of the value function.
>
> Question: “keep only the policies that achieve max performance....?”, we would like to refer the reviewer to the reformulation of reward minimization problem in equation 15. There you can see that for any feasible solution w, there is a soft inequality restricting all the policies to have a lower or equal value to that of the optimal policy. These constraints can be binding or not, but both cases are feasible. In fact, if the set of policies is diverse enough, the worst case reward will be chosen such that there is equality. For example think about the simple case where the SFs are: $(0,1)$ and $(1,0)$. The worst case reward will be $(\frac{-1}{\sqrt(2)}, \frac{-1}{\sqrt(2)})$, and the two policies will maximize it. For more general cases, we refer the reviewer to Figure 2, a, where we validate that empirically in DM control. The blue bars correspond to the number of active policies (that attain the max) after each iteration. The number of such policies is increasing and is clearly larger than 1.
>
> Question: “why there is not another w that makes this policy fail”, this is exactly what our proof shows. We would like to refer the reviewer to the definition of the features in the second paragraph of the preliminaries section. From there, it is clear that the features are always positive, and are for dimension d. The later answers the reviewer question regarding d in Lemma 3. The former is important for the proof of the uniqueness of the worst case reward. We hope that Lemma 5 in the revised version will answer the reviewer’s concerns. Note that the places that were changed are in blue color. We explain some parts clearer and more rigorously than in the previous version and we hope that you will find that satisfactory.
>
> Regarding the reward-set setting. Please note that we defined the features to be positive. Although the features are positive w can be negative and in general will be negative. So, for a given set of policies, the solution for the worst case reward is not just the min reward at each state. For example, if your set includes only the policy $(1,0)$ then the worst case reward will be $(-1,0)$ and not $(\frac{-1}{\sqrt(2)}, \frac{-1}{\sqrt(2)})$. We hope this addresses your concern regarding w_min. Regarding your question about the linearity in the features, see our answer in a separate post(note that this answer is also an answer to R1).
>
> “why optimize the policy set according to SMP?” This is a good question. The focus of this work is on the case that the reward is unknown. Therefore, if our set only includes a single policy, then the worst case reward w.r.t to it will always be “devastating”. Mathematically, this means that if the set of the policies is not diverse, then the worst case reward can choose to be minus the SFs of one of the policies (normalised), that is, to be as adversarial as possible w.r.t a single policy. When the set includes more than one policy, then the policies, under an SMP may complement each other. That is, if the reward is too adversarial w.r.t to a single policy, then it is likely that another policy in the set will be better w.r.t it. This is what happens in practice, when there is more than one policy that maximizes the worst case reward (active policies). In that case, the analytical solution of the worst case reward is not minus the SFs of one of the policies in the set. As a result, the value of the SMP w.r.t the worst case reward is better than that of that of the best policy in the set in isolation.
>
> “stick to the simplest setting of choosing the best policy” -- We revisited the problem formulation paragraph to make it clearer that we focus on the SMP as the mechanism that select policies. Please note that once algorithm 1 finishes and returns a set of policies, this set can be used by other SIPs (such as GPI) to yield better performance. We verify that empirically in Figure 1a.
>
> We hope that our response addressed all the reviewer’s concerns, but if it didn’t, please point us to the parts that we missed.

---

> > ### Author Response · Authors · 2020-11-13
> > **Linearity of the reward in the features**
> >
> > It is indeed true that for a *fixed* set of features the linearity assumption is restricting. However, it is this assumption that allows us to develop theory and thus provide theoretical guarantees for the proposed algorithm. Note that the guarantees we provide are in fact applicable in practice, as we illustrate in our experiments. For example, the experiments in the tabular MDP follow the theoretical framework exactly and achieve the upper bound that we developed (see more on that in our answer to point 2 below). The experiments in DM control are also very close to the theory, the only deviation is that we use DRL to optimize the policy for a given reward. We note that this agreement between theory and practice is in fact one of the strengths of our work, since more often than not there is a gap between the two.
> >
> > All that said, we point out that in the most general scenario we are free to define the features themselves. Note that, although the rewards are linear in the features, the features themselves can be arbitrary nonlinear functions of state-action pairs. This means that when we are able to define the features the linear assumption is in fact not so strong: for example, in [1], the authors discuss how in the tabular case this is not a restriction at all, and how in continuous state spaces we can find features that approximate any reward function with a certain accuracy.
> >
> > As a somewhat counterintuitive observation, we note that in many problems it is in fact easy to handcraft simple features that generate useful behaviour. This is illustrated in our experiments with DM control, in which using the standard features provided in the suite we were able to generate rich behaviour. For example, we were able to teach the walker to do salta jumps (see the video in the supplementary material) by simply using linear combinations of the standard features.
> >
> > Thirdly, our experiments suggest that the assumption is not too restricting in interesting problems. For example, we were able to teach the walker to do salta jumps (see the video in the supplementary material) under this assumption.
> >
> > Lastly, we believe that it should be easy to generalize our approach to a more general setup where the reward is represented as a nonlinear (perhaps a DNN) of the features. In this case, the minimization over w will not be  convex but will still be possible via the same techniques (SGD). This kind of algorithm will resemble GAIL (with a GAN) and we believe that it is an exciting direction of research for future work.
> >
> >
> > [1] Barreto, A., Hou, S., Borsa, D., Silver, D., & Precup, D. "Fast reinforcement learning with generalized policy updates." Proceedings of the National Academy of Sciences (2020).
> > [2] Abbeel, Pieter, and Andrew Y. Ng. "Apprenticeship learning via inverse reinforcement learning." Proceedings of the twenty-first international conference on Machine learning. 2004.
> > [3] Zahavy, Tom, Alon Cohen, Haim Kaplan, and Yishay Mansour. "Apprenticeship Learning via Frank-Wolfe." AAAI (2020).

---

> > > ### Comment · AnonReviewer2 · 2020-11-19
> > > **Focus on the ball reward definition**
> > >
> > > First of, sorry for the bad formatting, I fixed it. I've also noticed only now that we could use the latex math mode and I'll use it from now on. Finally, sorry for not having been able to answer before.
> > >
> > > Thank you for your clarifications. They addressed most of my point but I am still confused about my first one, which is also unfortunately my main concern. Let me first recall my initial questioning:
> > > - although the reward-set setting is quite general: within a ball in the feature space, it includes very unnatural cases that make the problem artificially too complex, in my opinion. Indeed, usually, one should consider that the worst case reward function $r_{min}$ in a reward function family R is the one that is minimal in every state-action-state : $r_{min}(s,a,s') = \inf_{r\in R} r(s,a,s')$. In this case, the solution to equation (7) is straightforwardly the single policy that optimizes the return on $r_{min}$. Please discuss more the interest of considering $r=\psi \dot w$ with w in a ball (it implies that if some feature takes sometimes positive and sometimes negative values, there is not clear $w_{min}$, and therefore no clear $r_{min}$).
> > >
> > > Now, indeed, I've been mistaken, because I was thinking about the $\ell_\infty$ ball (infinite norm), not the $\ell_2$ ball (Euclidean norm). Can we agree that, if we take the $\ell_\infty$ ball instead, then the solution is trivial? So, why use the $\ell_2$ ball? Is it closer to the model uncertainty we need to represent? If this is the case, what is the loss of using the $\ell_\infty$ ball instead?

---

> > > > ### Author Response · Authors · 2020-11-19
> > > > **Response to Reviewer 2, cont.**
> > > >
> > > > We would like to thank the reviewer for their response. The reviewer raised a good point. There are many reasons to choose the $\ell_2$ ball: it includes all the directions; the magnitude of the reward doesn’t change the optimal policy in tabular MDPs; in robust optimization it is used when the uncertainty about some parameter is Gaussian and consequently the scaled $\ell_2$ ball contains the true parameter with high probability; it is the standard assumption in related work and specifically in Apprenticeship Learning. We now highlight one more property that distinguishes the $\ell_2$ ball from the $\ell_\infty$ ball and is in particular relevant to the reviewer’s question.
> > > >
> > > > Implied from the reviewer’s response, is that a possible solution to Eq 7 is to solve the following problem:
> > > > $$\max_{\pi\in\Pi} \min_{w\in\mathcal{W}} \psi(\pi)\cdot w. $$
> > > >
> > > > This formulation is similar to Apprenticeship Learning (AL) without an expert, which is different from our approach that is hierarchical. We refer the reviewer to our latest response to Reviewer 3 for a discussion on the similarities between our approach and AL and to Sec B in the supplementary material for more details.
> > > > As the reviewer suggested, in the case where $\mathcal{W}$ is the $\ell_\infty$ ball, the internal minimization problem in the above has a single solution - a vector with -1 in all of its coordinates. However, with other norm balls the solution to the internal minimization problem is a function of the policy. In the case of the $\ell_2$ ball, it is the negative SFs normalized. This is important, since it clarifies that solving the min-max AL problem is not as easy and typically requires solving an MDP in each iteration (see the reference for AL above for more details).
> > > >
> > > > Now, the fact that the worst case reward is a function of the policy (or the policy set) forces it to make a tradeoff -- it has to “choose” the coordinates it “wants” to be more adversarial for. This tradeoff is what encourages the worst case reward to be diverse across iterations (w.r.t different sets) and as a result it induces a diverse set of policies. Diversity was an important goal in this work -- in addition to minimizing Equation 7, we were interested in the diversity of the  policies that result from that process.
> > > > Thank you for pointing this out. We hope that we answered your question. We will add this discussion when we introduce the $\ell_2$ ball in the paper.

---

> > > > > ### Comment · AnonReviewer2 · 2020-11-20
> > > > > **I've raised my score to accept**
> > > > >
> > > > > I'd like to thank the authors for the detailed explanations. They made good points on the interest of studying the $\ell_2$ ball rather than the $\ell_infty$ one. As a consequence, I recommend accepting the paper.

---

### Official Review · AnonReviewer4 · 2020-10-28
**Strong and non-trivial theoretical contributions, interesting empirical insight that connects directly to the theory**

**Rating:** 7
**Confidence:** 4

**Review:**

Summary: the authors propose to solve a family of related tasks with shared features and rewards that are linear in the features and equivalent up to scaling factor. The main contributions are as follows:
- a novel framework for analyzing a broad family of generalized policies (policies that are generalized to arbitrary rewards in the task space), including the concept of a set improving policy (SIP), and providing two practical examples that fit this definition, namely the worst case set max policy (SMP) and the well known and studied generalized policy iteration (GPI). It is shown that it is always better to use GPI over SMP, making it an instance of SIP.
- a novel iterative method for building a policy library for solving the worst-case reward, formulated as a convex optimization problem, along with policy improvement guarantees, an informed method for stopping the algorithm, and the ability to remove redundant policies (termed inactive policies)
- an empirical evaluation that connects the proposed method to learning a policy library with a diverse set of skills. The theoretical results are also validated experimentally, on a grid world example and control problems from Deepmind.

Pros:
- the work is of very high quality, all motivations seem sound and the theoretical results seem correct
- the idea of active task selection for building the policy library is very interesting, and it is surprising that this has not been considered within the framework of Barreto et al., 2017 so far
- the work could be of significance in the apprenticeship/curriculum/meta-RL community, and it is nice to see a more theoretical treatment of this topic

Questions:
- If my understanding is correct, the authors use the orthogonal and random basis to propose w at each iteration, but evaluate the resulting SMP policies with respect to the optimized rewards from (8). I am wondering if this is a fair evaluation for the baselines, given that the policies are always evaluated on $w_t^{SMP}$, or whether a new set of tasks (a proper "test" set) sampled from B (the standard ball) should be used to fairly compare (8) with the baselines? This would really test the generalization of the method on new instances as well, and is also often standard in the literature for evaluating the performance of a learning policy set. In other words, how robust is the resulting policy library to solving new task instances not previously seen before?
- Also, one thing that could explain the poor performance of the orthogonal baseline is that the reward seems to be quite sparse when most of the basis elements are set to zero (in the one-hot phi case, wouldn't they be almost always uninformative?) In this case, a more suitable baseline that directly targets diversity could be defined as finding the $w_1, w_2 \dots w_T$ such that their coverage of the task space is maximized under some prior belief over w (e.g. the standard ball). If I am not mistaken, this problem is similar to the maximum coverage or voronoi tessellation problem, which could be solved in advance and then deployed. (e.g. Arslan, 2016)
- Performing well relative to the worst-case performance seems reasonable so that the agent does not do poorly on any one task, but it could also be overly conservative. That is, could there be situations where optimizing the worst case leads to the agent not successfully completing the desired objective (e.g getting stuck on locally optimal solution)?
- at each iteration when the new optimal policy is learned with respect to $w_\Pi^{SMP}$, is the idea of SMP or GPI and previously learned policies used to help learn this new policy, or is it learned entirely from scratch (e.g. by simple epsilon-greedy)?

Minor comments:
- the legends in Figure 1a/b and the axis font in Figure 1c could be increased, same with Figure 2
- is the $\max_i$ necessary in equation (8)?

Overall, this works proposes a coherent theory for policy improvement, that also leads to useful implementation and interesting empirical insight (and cool visualizations). It can often be hard to obtain all of these at once.

Arslan, Omur, and Daniel E. Koditschek. "Voronoi-based coverage control of heterogeneous disk-shaped robots." 2016 IEEE International Conference on Robotics and Automation (ICRA). IEEE, 2016.

---

> ### Author Response · Authors · 2020-11-13
> **Response to Reviewer 4**
>
> We would like to thank the reviewer for they/there feedback.
>
> Question 1.
>
> The reviewer is correct: for the baselines, we add policies to the set by following the baseline rule (either by sampling a random reward and optimizing it or by adding an orthogonal reward and optimizing it). For each method (including ours and the baseline) we have a different set of policies $\Pi^t$ in each iteration. At iteration t we compute the worst-case reward w.r.t to SMP’s current set, $w^*(\Pi^t)$, and report the performance of all the methods under this reward ($w^*$ is computed through (8)). This means that the reported performance of SMP is the worst possible across all the tasks, while this is not necessarily true for the baselines (that is, there might be tasks different from $w^*$ in which their performance is worse).
>
> The reviewer’s suggestion of adding a “test set” of tasks is interesting. We will do our best to add this experiment to the paper before the end of the rebuttal phase, and will certainly have it in the final version of the paper. As noted above, if we replace the task $w^*$ used in our evaluation with any other task, the performance of our algorithm (SMP) will improve. So, for example, if we report the average performance of our algorithm on a “test set”, as suggested, this value will be greater than the one reported. This is not necessarily the case for the baselines. On the other hand, the performance of the baselines on the test set can in fact be better than SMP’s, since this is not what our algorithm is optimizing. The choice between maximizing the worst-case performance and the expected performance involves several interesting trade-offs; we elaborate on this point below.
>
> Question 2.
> This is also a very interesting suggestion, it would indeed be an interesting reference point. Such an algorithm should work well if we have some prior knowledge of the distribution of w or are able to sample from it (for example, by learning online as tasks are presented to the agent). However, we conjecture that the number of policies needed by such an approach to reach a good performance level would in general be considerably larger than the number of policies used by our algorithm (since in this case one has to cover all the support of the distribution over w with non-negligible probability mass).
>
> Optimizing for the worst case scenario gives us some benefits. First, we do not need to know anything about the distribution of w. This allows us to do things like building the library of policies in a completely unsupervised way, before ever seeing an actual task. Second, we can be very efficient: in our experiments we always found a set of diverse policies after only a few iterations. Focusing on the worst-case reward can also be very useful in scenarios where bad performance has a high cost associated with it: for example, if the agent is an autonomous vehicle, the priority might be to avoid accidents.
> All that said, we believe that the two approaches (optimizing for the worst-case or expected performance) are in fact complimentary. Finding a feasible way to cover the space of rewards and combine it with our approach is an exciting direction for future work.
>
> We will add the discussion above to the paper.
>
> Question 3.
> This is related to the previous question, and also an interesting point. We did observe in our experiments that sometimes our algorithm converged to the optimal value “too fast”. Concretely, this meant that after adding 2-4 policies to the set, newly-added  policies were not diverse or meaningful because w*, the worst-case reward w.r.t the SMP computed through (8), was very close to being a vector with -1 in all of its coordinates. That is, after a few iterations, the benefit of adding a new policy diminished quickly. One direction that we explored to alleviate this issue was to regularize the worst-case reward w* to have zero mean. Note that removing the mean does not change the task but potentially increases the difference in the relative magnitude of the entries in w*. This did indeed help in making the policies more diverse. These experiments have been added to the supplementary material (Section C).
>
> Question 4.
> We used a simple actor-critic agent with experience replay to learn each policy. We experienced both with the case where the parameters are learned from scratch and with the case where they are transferred from one task to another, but it did not seem to make a big difference (same for the experience replay). We did not use SMP or GPI to learn the new task, although this is a great idea to be explored in the future (we will mention it in the paper).

---

> > ### Comment · AnonReviewer4 · 2020-11-15
> > **Response to authors**
> >
> > Thank you very much for the clarifications, and for the updated paper and experiments. The authors' interpretation of risk/robustness as it relates to the framework is quite interesting. It could be interesting to see the emergence of "safer" behaviors of the agent on complex tasks such as driving, in the absence of the true reward, in future work. I am happy with the response and do not have further questions.

---

### Official Review · AnonReviewer3 · 2020-10-29
**The paper addresses an interesting problem, and the proposed approach is clear and well formalized. The paper analyzes the proposed approach theoretically as well as empirically, attaining good results.**

**Rating:** 8
**Confidence:** 3

**Review:**

= Overview =

The paper introduces an approach that, given a set of "basis" policies, constructs a high-level policy from the basis policies that is able to perform well in a variety of distinct (but related) tasks. Such tasks are described by MDPs with similar state-action spaces and similar dynamics, and differing only on the reward functions, all of which are built as a linear combination of common features.

Given a set of policies, the paper introduces the notion of "set improving policy" as a policy that outperforms any policy in the given set on the family of considered tasks. It provides two examples of such policies (SMP and GPI) and formalizes the problem of computing a SIP with maximal worst-case performance on the set of considered tasks as a max-min problem. It then contributes an incremental algorithm for this problem. The proposed approach is tested in a grid-world environment and the DM control suite.

= Positive points =

The paper is very well written, with the proposed approach clearly motivated, presented and analyzed. The proposed approach is novel, to the extent of my knowledge, and analyzed both theoretically and empirically.

= Negative points =

My main criticism is, perhaps, some lack of detail on the experimental evaluation -- particularly in the DM control suite.

= Comments =

Overall, I really enjoyed reading the paper. The problem addressed -- that of building a policy that performs well in a number of related tasks from a set of "simpler" policies -- is, in my view, quite relevant for the RL community, and has potentially interesting applications in domains such as robotics.

The proposed approach is, as far as I know, original and contributes to the state of the art. The paper briefly links its contributions to the existing literature on apprenticeship learning and hierarchical RL, but I would have appreciated some more discussion on these topics -- particularly, I'd like to better understand how the learned policy relates with policies taught through apprenticeship learning.

Overall, the ideas in the paper are presented in a very clear and elegant manner and the results strike me as technically sound. The proposed approach focuses on building a set of "basis" policies in such a way that the policy built from them performs as well as possible in all the considered family of tasks. The method is derived from first principles, and the performance bounds provided (framed in terms of the performance of the SMP policy) are then validated empirically.

Finally, the paper is evaluated in a smaller grid-world domain and in the DM control suite. One aspect that could, perhaps, be improved is concerned with the description of the empirical evaluation in the DM control suite: the paper does describe how the family of rewards for these tasks were built, but it would be good to provide some description of the types of policies that the different rewards lead to and how the policy computed by the proposed approach relates (or differs) from them.

---

> ### Author Response · Authors · 2020-11-13
> **Response to Reviewer 3**
>
> We would like to thank the reviewer for they/there feedback. The reviewer raised a question about the policies learned by following our algorithm: how do they relate to Apprenticeship Learning (AL), and can we better show if they learned similar/different things from one another.
>
> Relation to AL. This is a great question and we debated about it when working on this paper. Both AL and our algorithm can be used to solve the same goal: achieve good performance w.r.t the worst case reward. However, AL is concerned with finding a single policy, while our algorithm is explicitly designed to find a set of policies. To be more specific, we need to refer to a specific AL algorithm, so we will refer to the projection algorithm by Abbeel & Ng (04). The basic idea in this algorithm is that given an estimate of the SFs of the expert (we use our notation though in their paper they refer to a similar quantity as feature expectations) we want to find its projection onto the SFs polytope as well as a policy whose SFs equal this projection. We refer the reviewer to Sec B in the supplementary material for more details, but provide a short summary here for the following discussion. The algorithm achieves that by maintaining a set of policies, and  convex combination of coefficients over the set of policies, known as a mixed policy, such that its SFs are the convex combination of the SFs in the set. The goal is to find the convex combination whose corresponding SFs are closest to those of the expert in the L2 norm. In each iteration, a policy is added to the set by maximizing a reward signal that is defined to be the negative gradient of this objective.
>
> There is no direct connection between the policies that are discovered from following these two processes. This is because the intrinsic rewards that are maximised by each algorithm are essentially different. Another way to think about this is that since the policy that is returned by AL is a mixed policy, its goal is to return a set of policies that are similar to the expert, but not diverse from one another. From a geometric perspective, the policies returned by AL are the nodes of the face in the polytope that is closest to the demonstrated SFs. Even more concretely, if the SFs of the expert are given exactly (instead of being approximated from trajectories), then the AL algorithm would return a single vertex (policy). Finally, while a mixed policy can be viewed as a composition of policies, it is not a SIP. Therefore, it does not encourage diversity in the set. Our algorithm, on the other hand, is explicitly designed to return a diverse set of policies.
>
> “Description of the types of policies that the different rewards lead to and how the policy computed by the proposed approach relates (or differs) from them”. We would like to refer the reviewer to Figure 3, where we visualize the policies learned by our algorithm in DM control. There we show a snippet of the trajectories taken by different policies. In the supplementary material, we further provided videos that we recorded from these policies. So, for example, the pendulum balances itself up and down, and the cheetah tries to stand on either leg, or to walk in either direction. The walker and the hopper discovered other locomotion skills that are not expected, but the key finding is that they are indeed very diverse from one another. We hope that this is what the reviewer asked for, but in case the reviewer believes that there are some missing details, or in case that we didn’t answer all of they/there questions, please let us know what you think is missing and we would provide more details.

---

> > ### Comment · AnonReviewer3 · 2020-11-15
> > **Response to authors**
> >
> > I would like to thank the authors for the clarifications. I am happy with the response, as I believe it addresses the issues that I raised in my review.

---

### Author Response · Authors · 2020-11-13
**Revised version**

Dear reviewers, we have uploaded a revised version of our paper to reflect your comments. For your convenience, most of the changes are marked in blue color. Smaller notation changes were also fixed.

---

### Author Response · Authors · 2020-11-13
**Experiments on a test set of rewards**

Dear reviewers, we would like to thank Reviewers 1 and 4, for expressing interest in the performance of our algorithm on a test set of rewards. This is an interesting setup which we didn't consider when we submitted our paper. We have now uploaded a new version of the paper where we performed this experiment in the supplementary material Section D. We hope that this is what you referred to in your review, but if it isn't please let us know.

For your convenience, we also provide a short description of the experiment here. We trained our algorithm and the two baselines in the same manner as we did before. During evaluation, we tested the performance of each method on a holdout set of rewards that were sampled uniformly over the unit ball. The results suggest that our algorithm achieves better performance than the two baselines when measured on this set of unseen rewards. More importantly, to achieve the same level of performance, our algorithm requires significantly fewer policies than the baselines. For more details, please refer to the Supplementary D.

---

### Decision · Program_Chairs · 2021-01-07
**Final Decision**

**Decision:**

Accept (Spotlight)

**Comment:**

All reviewers are positive or very positive about this work. The authors successfully addressed all questions. I believe this paper should be accepted.